# Cytoskeletal dynamics regulates stromal invasion behavior of distinct liver cancer subtypes

Ryan Y. Nguyen [1,5], Hugh Xiao[1,5], Xiangyu Gong [1,5], Alfredo Arroyo[2], Aidan T. Cabral[1], Tom T. Fischer [2,3], Kaitlin M. Flores[1], Xuchen Zhang[4], Marie E. Robert[4], Barbara E. Ehrlich [2] & Michael Mak [1✉]

Drug treatment against liver cancer has limited efficacy due to heterogeneous response among liver cancer subtypes. In addition, the functional biophysical phenotypes which arise from this heterogeneity and contribute to aggressive invasive behavior remain poorly understood. This study interrogated how heterogeneity in liver cancer subtypes contributes to differences in invasive phenotypes and drug response. Utilizing histological analysis, quantitative 2D invasion metrics, reconstituted 3D hydrogels, and bioinformatics, our study linked cytoskeletal dynamics to differential invasion profiles and drug resistance in liver cancer subtypes. We investigated cytoskeletal regulation in 2D and 3D culture environments using two liver cancer cell lines, SNU-475 and HepG2, chosen for their distinct cytoskeletal features and invasion profiles. For SNU-475 cells, a model for aggressive liver cancer, many cytoskeletal inhibitors abrogated 2D migration but only some suppressed 3D migration. For HepG2 cells, cytoskeletal inhibition did not significantly affect 3D migration but did affect proliferative capabilities and spheroid core growth. This study highlights cytoskeleton driven phenotypic variation, their consequences and coexistence within the same tumor, as well as efficacy of targeting biophysical phenotypes that may be masked in traditional screens against tumor growth.

[1] Department of Biomedical Engineering, Yale University, New Haven, CT, USA. [2] Department of Pharmacology, Yale University, New Haven, CT, USA. [3] Institute of Pharmacology, University of Heidelberg, Heidelberg, Germany. [4] Department of Pathology, Yale University, New Haven, CT, USA. [5]These authors contributed equally: Ryan Y. Nguyen, Hugh Xiao, Xiangyu Gong. ✉email: michael.mak@yale.edu

Liver cancer is the fourth leading cause of cancer-related deaths and exhibits a wide variety of genotypic and phenotypic variation[1]. Within the same tumor nodule, liver cancer can present with a variety of unique morphologies, growth patterns, and migration profiles[2]. As with other cancers, such heterogeneity confers constant tumor evolution and, subsequently, can result in the ability of liver cancer cells to adapt to pharmacological pressures[1,3]. Although advances in genomics and pathology have made efforts to pinpoint the molecular mechanisms behind this heterogeneous drug resistance, the functional mechanical and biophysical phenotypes, like proliferation and invasion, which result from this heterogeneity remain poorly understood[4]. Without an understanding of how to effectively treat the phenotypic outputs resulting from this heterogeneity, there remains a gap as to how to assign effective drug targets to each of these subtypes. Thus, being able to relate heterogeneity to dynamic functional readouts such as proliferation and invasion is crucial in understanding the progression of the disease and for developing drug therapies that effectively target these subpopulations.

The ability for cancer cells to proliferate and invade is largely regulated by the cytoskeleton, which controls many aspects of cellular function, including cell shape, protrusion dynamics, and migration[5]. The cytoskeleton is regulated by a multitude of molecular and mechanical components, which work together to propel the cell forward. Three of the main signals which contribute to cell motility are lamellipodia formation, actin turnover, and cell contractility. The main force that powers the outward projection on the leading edge is the polymerization of actin which branches and extends to push forward lamellipodia which are flat, broad membrane protrusions[6]. This leading-edge projection is largely modulated by the rate of actin turnover at the leading edge[7]. Simultaneously, adhesions at the rear-end of the cell detach by rear-end actomyosin contractility and the disassembly of actin[8,9]. Together, the interplay of leading-edge pushing and trailing edge retracting drives cell migration. At the core of this interplay is the Rho family of small GTPases. The key members of this family include RhoA which controls actin contractility, Rac1 which controls lamellipodia formation, and Cdc42, which is involved in filopodia formation[10].

Both Rac1 and RhoA have been implicated in invasive behavior of various tumor types including breast, lung, colon, and liver cancer. Dysregulation of activity of both Rac1 and RhoA has been linked to mesenchymal tumor movement and invasive phenotypes in cancer[11]. In addition, their downstream effectors have also been implicated in generating invasive cell motility. Regulators that affect actin turnover at the leading edge such as cofilin, which disassembles actin filaments[12], and regulators that affect rear-end actomyosin contractility, such as Rho-associated Protein Kinase (ROCK)[13], also play key roles in cell migration. Moreover, actin turnover and contractility allow for cells to generate forces to physically interact with the fibrillar extracellular matrix (ECM) and facilitate invasion. Studies have linked contractility and actin turnover to aberrant physical interactions with the ECM and greater metastatic and invasive capability in many cancers including liver cancer[14–17]. Although components of the Rho family of small GTPases have become key drug targets for cancer therapeutics, cancer cells display heterogeneous cytoskeletal regulation and have differing responses to drug treatment[18].

Traditional drug discovery in cancer therapy has largely focused on the bulk killing of tumors, and previous studies of liver cancer have identified key somatic mutations that dictate heterogeneous drug response[19]. However, many of these studies focus mainly on static readouts such as live/dead assays and histological analysis and do not capture key dynamic functional behaviors which result from this heterogeneity. In addition,

in vivo studies tend to look at endpoint analysis such as bulk tumor growth, which does not fully capture the intermediate progression and heterogeneous subtype evolution of the disease. Endpoint analysis also does not indicate responses of cancer subtypes, which may reveal several important drivers of cancer progression. Furthermore, these studies do not fully capture the complex physical tumor environment. Cells being studied in these screens are cultured on 2D glass or plastic substrates, which have an effective stiffness of $10^9$ Pa whereas the stiffness of a tumor stroma is ~$10^3$ Pa[14]. When compared to 2D culture environments, 3D environments produce tumor growth, invasion, and drug resistance profiles which more closely resemble those seen in patients and therefore represent a clinically more relevant platform for drug testing[20–22]. 3D hydrogel systems such as those made of collagen capture the importance of fibrillar architecture and provide a more physiologically relevant setting for interrogating liver cancer cell dynamics. These differences attributed to cell-ECM interactions severely alter liver cancer cell response to drugs[14,21]. Recent studies have been able to encapsulate liver cancer cells on decellularized liver matrices in an attempt to understand cancer cell proliferation dynamics; however, these studies focus on biochemical regulation of liver cancer cells with less attention given to invasive biophysical and cytoskeletal interactions with the matrix[23,24]. Assessments of 3D interactions with the ECM in conjunction with standard readouts of drug response in cancer are needed to understand how cytoskeletal inhibition differentially affects invasion dynamics of different liver cancer subtypes.

To this end, our study takes an integrative histological, biophysical, and bioinformatics approach to explore the role of cytoskeletal regulation of the phenotypic diversity in liver cancer. We compared the morpho-dynamics and invasion profiles of two distinct liver cancer cell lines: SNU-475 cells which are more mesenchymal-like and HepG2 cells which are more hepatoblast-like[19]. We integrated our findings from our cell line studies with tissue samples from human patients with liver cancer. These patient-derived tissue samples contained regions with tumor cell phenotypes that are comparable to both cell lines tested. To examine the distinct morpho-molecular dynamics of these lines, we performed quantitative immunofluorescence imaging examining actin architecture and cell shape in conjunction with immunoblotting. We applied 2D scratch assays and 3D spheroid invasion assays to investigate dynamic functional readouts of cell invasion of our cell lines. When using these assays, we also utilized component-specific drugs to target different aspects of cytoskeletal regulation. Our study revealed that inhibition in 2D invasion does not necessarily translate to inhibited migration in 3D. We found that cytoskeletal targeting impacts different cancer cell subtypes differently, with more drastic effects on the more rapidly migrating SNU-475 line compared to the more rapidly growing HepG2 line. To further expound our findings to a clinical population level, we performed bioinformatics analysis on the molecular signatures of hepatocellular carcinoma (HCC) patients from the Cancer Genome Atlas Program (TCGA) database. In agreement with our biochemical and biophysical results, we found that components of the cytoskeletal machinery tested were significantly upregulated in HCC patients and correlated with worse survival. Overall, our study demonstrated biophysical phenotypic diversity in liver tumors, with an emphasis on dynamic behaviors, cytoskeletal regulation, therapeutic susceptibility, and clinical relevance.

## Results

**Liver cancers display a multitude of distinct invasive behaviors.**
Liver cancer architecture was assessed by histological examination

of patient-derived samples with background liver, HCC confined within a fibrotic capsule, and HCC invading into the fibrotic stroma (Fig. 1a). In the case of HCC encapsulated by a fibrotic capsule, liver cancer cells expanded by collectively growing against the surrounding environment (Fig. 1a, iv-v). HCC could also interact with the surrounding stroma with discrete HCC cells invading into the fibrotic stroma (Fig. 1a, vii-viii). Moreover, tumor cells confined by the surrounding fibrotic capsule and those which invade into the stroma could both be found within the same tumor (Supplementary Fig. 1). We performed E-cadherin IHC on our tissue samples as an epithelial cell indicator (Fig. 1a, iii, vi, ix). Liver cancer cells confined within the fibrotic capsule showed higher levels of E-cadherin expression while the stromally invasive liver cancer cells had less E-cadherin expression. This further indicated the presence of two distinct liver cancer cell subtypes with the encapsulated liver cancer being more epithelial and the stromally invasive cancer being less epithelial. These two cancer cell subpopulations have been reported in other studies and may have the diverse functional capability and subsequently therapeutic response[25]. We developed a simple hypothetical schematic to highlight the distinct features of these two subtypes (Fig. 1b).

In one case, HCC can be confined by the matrix (Fig. 1b, left). As such, this subtype displays less local dissemination and is surrounded by a fibrotic stroma. This fibrotic confinement has been observed in a variety of solid tumor types like HCC, neuroendocrine tumors, renal cell carcinoma, and adrenal cortical carcinoma[2,26]. Invasive cells in solid tumors can also migrate into the surrounding stroma. Other studies have found that invasive cells are often highly active and can send out dynamic protrusions which exert contractile forces on collagen to allow alignment and densification which facilitates further invasion (Fig. 1b, right)[9,15,27]. This hypothesis of collagen remodeling facilitating subsequent invasion is consistent with other tumor types including breast, lung, and colorectal cancer[28–30]. These studies have shown that invasive cells have distinct cytoskeletal signatures that enable invasion and cell-matrix interactions. By looking at cell lines that are representative of these subpopulations, we sought to understand the mechanobiological differences between liver cancer subpopulations and how their distinct cytoskeletal features play a role in invasive behavior.

**SNU-475 and HepG2 cells have distinct biochemical, physical, and invasive signatures.** To study the differential responses of liver cancer cells in 2D and 3D environments, we compared the SNU-475 cell line, which is more invasive and mesenchymal, and the HepG2 cell line, which is less invasive and more epithelial[19]. We first assessed the differences between these two cell lines on 2D collagen-coated surfaces. Collagen was chosen because it is the main component of the ECM and is a major ligand for integrin and subsequent cytoskeletal machinery in cell migration[31]. Fluorescent staining of both lines showed that SNU-475 cells had pronounced actin stress fibers and had an elongated cellular morphology, whereas HepG2 cells did not have these stress fibers and tended to be rounded and clump together (Fig. 2a). Such morphology led to decreased overall cell area in HepG2 cells compared to SNU-475 cells. Immunoblotting indicated that HepG2 cells have higher levels of E-cadherin than do SNU-475 cells (Fig. 2b, Supplementary Fig. 2). Furthermore, HepG2 cells had significantly lower levels of phosphorylated cofilin (p-cofilin) and phosphorylated myosin II light chain (p-MLC) relative to SNU-475 cells (Fig. 2b), both of which are key signaling molecules in actin turnover and have been implicated in metastatic behaviors[32]. The high levels of E-cadherin in HepG2 cells showed that these cells are more epithelial-like and suggested that the HepG2 cells have less migratory capability than SNU-475 cells[33]. We next performed

scratch assays on collagen-coated surfaces to observe wound closure rate (Fig. 2c). In 48 hours, SNU-475 cells were able to fully close the wound, but HepG2 cells were only able to close the wound 50% (Fig. 2d). HepG2 cells require up to ~144 hours to close wounds after making the scratch. In conjunction with our histological analysis, these results demonstrated that our cell lines tested are representative of distinct cancer subtypes seen in vivo.

To better understand liver cancer cell motility and tumor aggressiveness in a more in vivo-like setting, we grew SNU-475 and HepG2 cells into spheroids and encapsulated them in 3D collagen hydrogels. The escape of cells out of these spheroids was monitored over 5 days (Fig. 2e). SNU-475 spheroids tended to form spherical shapes, whereas HepG2 spheroids tended to be flat and spread out. As described previously[34], HepG2 cells doubled more quickly than SNU-475 cells (48 h vs. 66 h), explaining differences in final spheroid size. Qualitative assessment of collagen matrix remodelling was observed by brighter reflectance signals which have been used previously to describe collagen remodeling (Fig. 2e)[35]. SNU-475 cells tended to detach from the spheroid and migrated into the surrounding gel more than HepG2 cells (Fig. 2f, i-ii). After cells escaped from the spheroid, SNU-475 cells migrated much farther than HepG2 cells (Fig. 2f, iii-iv). SNU-475 spheroids caused more matrix remodeling than HepG2 spheroids (Fig. 2f, v). HepG2 cells tended not to escape out of the spheroid, resulting in an overall expansion of the HepG2 spheroid that was greater than SNU-475 spheroid expansion (Fig. 2f, vi). Overall, these comparisons highlight the stark differences in SNU-475 and HepG2 cells molecularly and in motility in both 2D and 3D.

**SNU-475 and HepG2 cell lines have distinct 2D responses to cytoskeletal inhibitors.** To identify pharmacological means to alter 2D cell migration in liver cancer, we targeted key regulators of the Rho GTPases, including calcium signaling, Rac1, PAK4, LIMK1, MLCK, and actomyosin contractility (Fig. 3a). Scratch assays over 48 h (SNU-475 cells) or 144 h (HepG2 cells) allowed assessment of wound closure under drug treatment (Fig. 3b, Supplementary Fig. 3). For both cell lines, we also performed scratch assays during Sorafenib treatment, which is the most commonly used treatment for HCC[36]. HepG2 cells treated with Sorafenib had low viability and did not close the wound. In addition to treatment with single drugs, we also used a double drug treatment to test for synergy among pathways. In particular, we tested actomyosin contractility and LIMK1 inhibition co-treatment ((S)-4′-nitro blebbistatin (blebbistatin) + LIMKi3, respectively). This combination allowed assessment of the effects of PAK4, which is able to phosphorylate LIMK1 and, in turn, phosphorylate cofilin to its inactive form as well as control actomyosin contractility via the Rho-ROCK signaling axis[37]. For the SNU-475 cells, all drugs except for calcium inhibition via BAPTA were able to significantly inhibit wound closure. To interrogate cytoskeletal machinery that is directly downstream of calcium signaling, we tested inhibition of the calcium-dependent myosin light chain kinase (MLCK). We found a significant decrease in wound closure for both cell lines after MLCK inhibition (Fig. 3b). The blebbistatin-LIMKi3-cotreated SNU-475 cells achieved wound closure to levels comparable to the blebbistatin treatment alone (Fig. 3b, left panel). For the HepG2 cells, however, only inhibition of PAK4, LIMK1, and dual inhibition of LIMK1 and actomyosin contractility significantly inhibited wound closure (Fig. 3b, right panel). Taken together, these data suggest that SNU-475 cells are more sensitive to different cytoskeletal perturbations than HepG2 cells.

To better understand how these drugs inhibit migratory phenotypes, quantitative immunofluorescence was monitored to

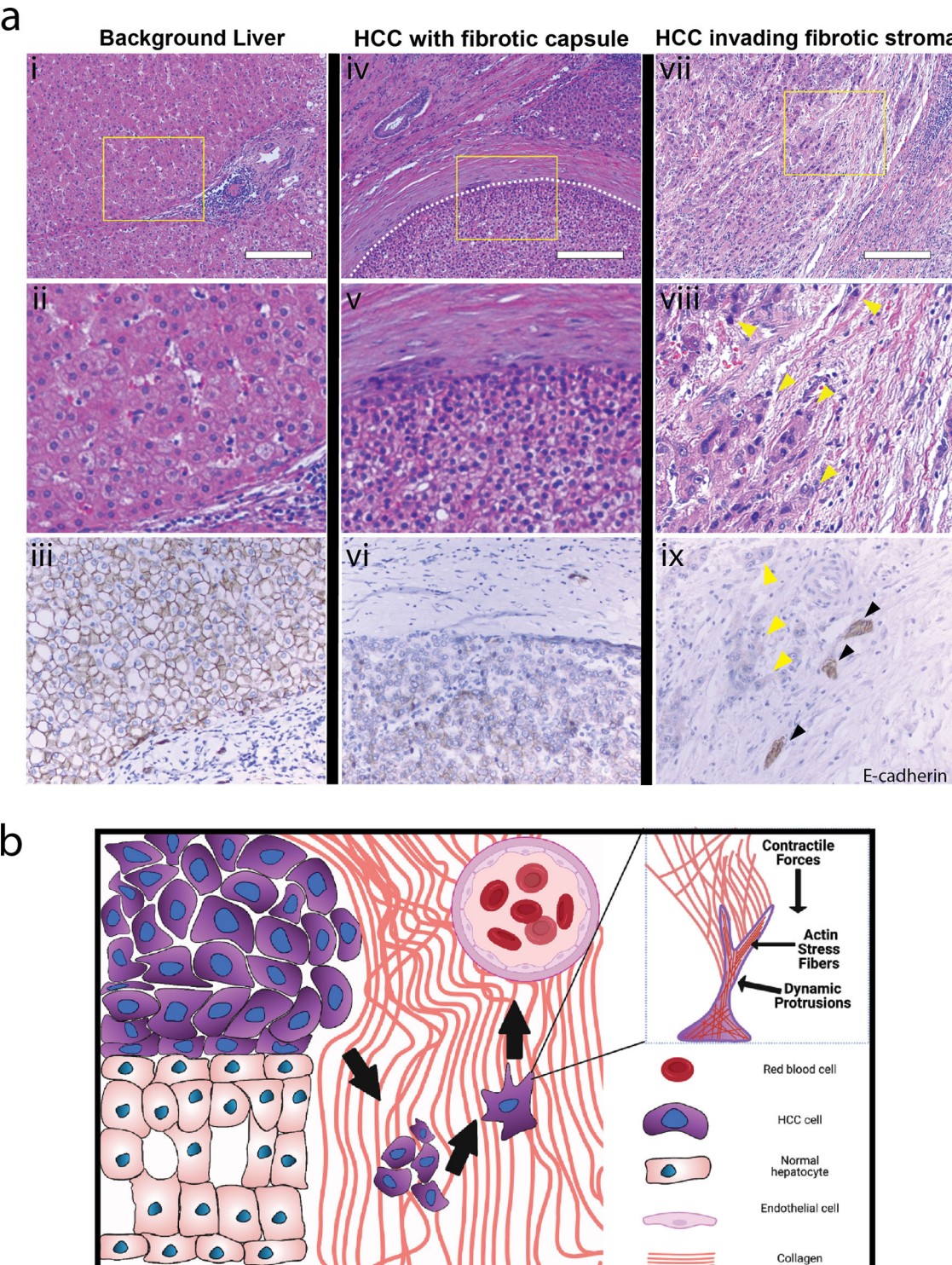

**Fig. 1 Distinct morphological differences and invasive modes of HCC. a** Background liver shows well ordered liver lobule and portal tract with mild chronic inflammation, portal and periportal fibrosis (i–iii), HCC encapsulated by a fibrotic capsule (iv–vi), and HCC tumor cells invading into the fibrotic stroma (vii–ix). Samples are hematoxylin and eosin stained. Lower panels show IHC staining of E-cadherin in approximately corresponding regions. Scale bars: 200 μm. Yellow arrows indicate stromal invasion of HCC in collagen while black arrows indicate bile ductules. **b** Our working model illustrates invasion modes of HCC. While HCC tumor cells can proliferate against the fibrotic capsule, HCC tumor cells can also invade into the fibrotic stroma. HCC tumor cells can send out dynamic protrusions and exert contractile forces on the collagen to allow for alignment and densification which will facilitate further invasion. Made with Biorender.com.

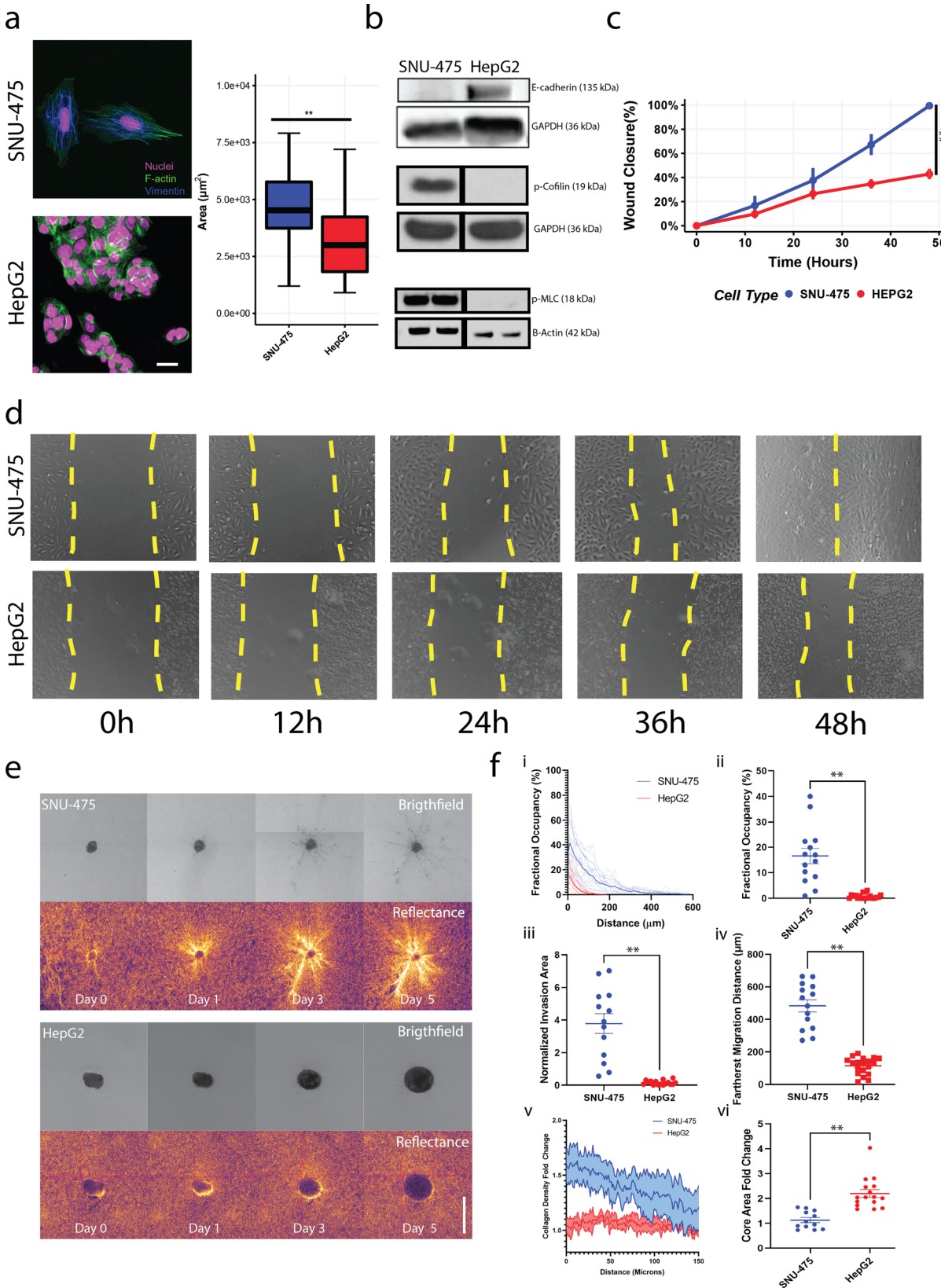

determine effects on cell morphology. We quantified nuclear aspect ratio, nuclear circularity, cell aspect ratio, cell circularity, cell area, and cellular solidity (Fig. 3c, Supplementary Figs. 4-5). Both PAK4 and LIMK1 inhibition decreased cell spreading area in SNU-475 cells significantly (Fig. 3d, i). In contrast, PAK4 inhibition, but not LIMK1 inhibition, was able to significantly increase cell circularity and decrease HepG2 cell spreading (Fig. 3d, ii). Although LIMK1 appeared to inhibit HepG2 wound closure rate, it did not affect HepG2 nuclear or cellular morphology. Interestingly, PAK4 inhibition caused HepG2 cells to have increased solidity (Fig. 3d, ii). This was a result of the decrease in the spread area as well as a decrease in lamellipodial

**Fig. 2 Molecular and physical differences between SNU-475 and HepG2 cells. a** Representative immunofluorescent images of SNU-475 cells and HepG2 cells. Green is F-actin, magenta is DAPI, and blue is Vimentin. Scale bar is 25 μm. **b** Immunoblotting of SNU-475 and HepG2 cells for E-cadherin, p-cofilin, and p-MLC. Black vertical separation lines for p-cofilin and p-MLC blot indicate that measurements are separated lanes of the same blot. **c** Scratch assay analyses and **d** corresponding representative images of scratch assays for SNU-475 and HepG2 wounds over 48 h. Yellow dotted lines indicate wound area. Single dotted line indicates that the wound is closed. $N = 11$ replicates for each cell line. **e** SNU-475 (top panel) and HepG2 spheroids (bottom panel) embedded in collagen matrices and imaged over 5 days. Images depict brightfield and reflectance microscopy of each spheroid. Scale bar: 500 μm. **f** After 5 days of culture in collagen gel, the percentage occupancy of the disseminated cells at each of the distances from the spheroid periphery is calculated (i) and the percentage occupancy at 100 μm away from the spheroid periphery are compared between the two cell lines (ii). The total areas of the disseminated cells after normalization (iii) and the farthest cell migration distances (iv) are compared between the two cell lines. The collagen densification as a function of distance from the spheroid (v) and core area fold change are compared across cell lines (vi). Plots show mean ± SEM. $N = 14$ spheroids for each cell line. Unpaired $t$-test with Welch's correction is performed. *$P \leq 0.05$, **$P \leq 0.01$.

size. Blebbistatin treatment significantly decreased cell circularity, cell area, and cell solidity for SNU-475 cells. Co-treatment with LIMKi3 further exacerbated this effect (Fig. 3d, i). There was no significant change in nuclear aspect ratio or circularity for blebbistatin or blebbistatin + LIMKi3 treated HepG2 cells (Supplementary Fig. 5e-f). There was a decrease in HepG2 cell circularity with 5 or 10 μM blebbistatin, or 5 μM blebbistatin + 10 μM LIMKi3 treatment (Fig. 3d, ii). This decrease in HepG2 cell solidity was observed with either 10 μM blebbistatin treatment or 5 μM blebbistatin + 5 μM LIMKi3 treatment (Fig. 3d, ii). MLCK inhibition via ML-7 treatment decreased cellular solidity and circularity in SNU-475 cells (Fig. 3d, i, Supplementary Fig. 4d). Interestingly, ML-7 treatment decreased HepG2 spread area and subsequently caused an increase in cell circularity (Fig. 3d, ii, Supplementary Fig. 5b). To verify the effect of these drugs, we performed Western blotting after drug treatment (Fig. 3e, Supplementary Figs. 6-9). LIMKi3 treatment was able to significantly decrease p-MLC expression levels in SNU-475 cells while KPT treatment showed a trend towards decreasing p-MLC levels in SNU-475 cells. We also found decreasing trends in p-cofilin after KPT or LIMKi3 treatment for HepG2 cells (Fig. 3e). Overall, these data suggest that inhibition of actomyosin contractility will decrease 2D cell migration more than cofilin-mediated actin turnover in SNU-475 cells. Conversely, our data also suggest that HepG2 2D cell migration is more dependent on cofilin-mediated actin turnover than actomyosin contractility.

**Actin and lamellipodial dynamics are regulated differently in SNU-475 vs. HepG2 cells.** Based on the discrete interactions of each drug on cellular shape, we wondered if these shape changes were a result of changes in cellular actin organization or structure. We measured actin coherency (degree of alignment of actin stress fibers), actin intensity, and F-actin to cell area ratio. For the SNU-475 cells, actin coherency was reduced when cells were treated with LIMKi3, blebbistatin, the co-treatment of the two, or ML-7 (Fig. 4a). With the exception of Rac1 inhibition via NSC23766 treatment and MLCK inhibition via ML-7, all drugs were able to decrease actin intensity. With the exception of BAPTA and NSC23766 treatment, all drugs are able to decrease the F-actin area to cell area ratio. For the HepG2 cells, there were very few or no actin stress fibers, so there are no measures of actin coherency available. All of the drug treatments tested significantly changed HepG2 actin intensity and F-actin area to cell area ratio (Fig. 4b).

Due to the interplay of PAK4 and Rac1, we investigated the role of these two pathways in lamellipodial formation[38]. Consistent with previous studies, we found that Rac1 inhibition was sufficient to show decreasing lamellipodial number and to significantly decrease lamellipodial size in SNU-475 cells (Fig. 4c-d). PAK4 inhibition did not significantly affect lamellipodial size (Fig. 4d). Interestingly, HepG2 lamellipodia formation was not affected by Rac1 inhibition (Fig. 4e-f). PAK4 inhibition in HepG2 did not cause a decrease in lamellipodia formation but did

significantly decrease lamellipodia area (Fig. 4f). PAK4 also decreased HepG2 cell area (Fig. 4f). Taken together, overall HepG2 cell spreading and subsequent lamellipodial formation was more dependent on PAK4 than on Rac1.

**3D invasion dynamics of liver cancer spheroids is distinct from 2D invasion profiles.** Given the striking effects of cytoskeletal perturbation on 2D cell culture molecularly and morphologically, we next investigated how these perturbations would affect 3D liver cancer cell migration in collagen hydrogels. Before embedding either SNU-475 or HepG2 spheroids in collagen gels, we pretreated spheroids for 2 h with each drug; during the course of the experiment, media with appropriate drug concentration was replenished every other day to ensure a drug response over the entire course of the experiment (Fig. 5a). All spheroids were made with 1000 cells/ spheroid. Measures of the invasion area, farthest migration distance, percentage occupancy, and core area fold change were measured after 5 days of growth (Fig. 5b, Supplementary Fig. 10). We also measured changes in collagen densification as a result of drug treatment (Fig. 5d-e, Supplementary Fig. 11). For reference to well-studied treatments, we included SNU-475 spheroids treated with Sorafenib. Although high concentrations of Sorafenib (10 μM) were able to completely inhibit invasion in 2D and 3D, intermediate concentrations of Sorafenib (3 μM) did not inhibit 2D or 3D invasion significantly (Supplementary Fig. 12). To determine if the long-term culture of these spheroids with drugs would affect proliferation, we stained our spheroids with Ki67 after 5 days of culture and did not see any significant changes in expression (Supplementary Fig. 13).

BAPTA treatment was used to test the effects of calcium chelation on spheroid invasion. Although BAPTA treatment did not reduce 2D wound closure in SNU-475 cells, it was able to decrease SNU-475 spheroid invasion area and farthest migration distance (Fig. 5b). Furthermore, BAPTA treatment inhibited collagen densification and alignment. BAPTA treatment did not affect HepG2 spheroid invasion or collagen densification (Fig. 5c). Previously, BAPTA was reported to reduce collagen contraction[39].

We next tested the role of Rac1 inhibition on spheroid invasion. In agreement with our 2D data, SNU-475 spheroids treated to inhibit Rac1 have significantly decreased cell migration distance (Fig. 5b). Strikingly, Rac1- inhibited SNU-475 cells that have migrated out of the spheroid are not spindle-shaped, but instead adopt a circular shape (Fig. 5a, inset 3rd column). In addition, they tended to have less long-range collagen alignment and densification relative to control but were still able to generate short-range collagen alignment (Fig. 5d). Rac1 treatment neither affects HepG2 spheroid invasion nor collagen densification (Fig. 5d). It was previously shown that Rac1 is able to direct dynamic protrusion generation as well as ECM alignment in metastatic breast cancer cells[17]. Our results suggest Rac1 plays a similar role in HCC.

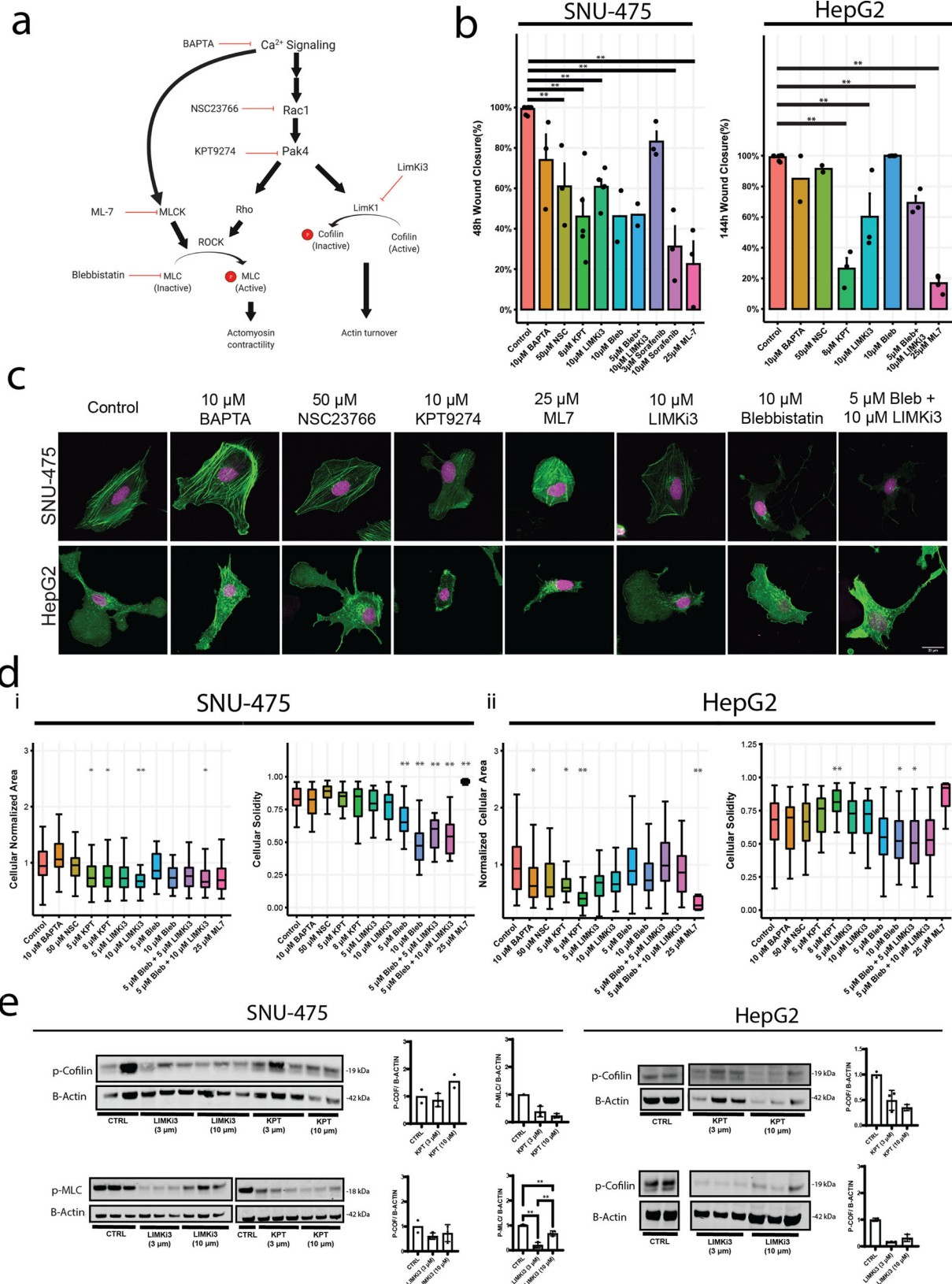

Due to the importance of PAK4 in 2D migration and morphology in both cell lines, we inhibited PAK4 with KPT9274 to test the role of PAK4 on spheroid invasion (Supplementary Fig. 14). PAK4 inhibition decreased the escape of SNU-475 cells out of the spheroid but did not affect spheroid core growth (Fig. 5b). Interestingly, PAK4 inhibition decreased HepG2 spheroid core growth but did not significantly affect escape into the surrounding matrix (Fig. 5b). When examining the surrounding matrix, KPT9274 reduced short-range and long-range collagen densification by SNU-475 spheroids (Fig. 5c-e, Supplementary Fig. 11). This suggested that PAK4 plays a role in SNU-475 matrix densification and alignment as well as invasion

**Fig. 3 2D Molecular and migratory phenotype profiling of SNU-475 and HepG2 cells. a** Pathway diagram illustrating the effect of each drug used on cytoskeletal machinery. **b** Scratch assay bar plots for SNU-475 and HepG2 for all drugs tested. Scratch assay bar plots at 48 h and 144 h for all drugs tested for SNU-475 and HepG2 cells, respectively. $N \geq 2$ replicates. Plots show mean ± SEM with conditions where $N > 2$. **c** Representative immunofluorescence images of SNU-475 spheroids (first row) and HepG2 spheroids (second row) treated with BAPTA (Ca2+ chelator), NSC23766 (Rac1 inhibitor), KPT9274 (PAK4 inhibitor), LIMKi3 (LIMK inhibitor), blebbistatin (myosin inhibitor inhibitor), both blebbistatin and LIMKi3, aor ML-7 (MLCK inhibitor), respectively. Green represents F-actin and magenta is DAPI. Green represents F-actin and magenta is DAPI. Scale bar: 30 μm. **d** SNU-475 (i) and HepG2 (ii) cellular area and solidity. $n > 30$ cells for each condition from $N = 2$ independent experiments. **e** p-MLC western blots for drug conditions for SNU-475 and HepG2 cells. Plots show mean ± SD. One-way ANOVA with Tukey post-hoc testing was performed. Significant difference ($p < 0.05$) was detected between any two of the conditions. *$P < 0.05$, **$P < 0.01$.

machinery in 3D cell migration. Because HepG2 spheroids do not typically cause obvious collagen densification, PAK4 inhibition did not affect collagen densification (Fig. 5d-e, Supplementary Fig. 15). Because collagen densification by HepG2 spheroids was likely caused by expansion against the surrounding collagen, PAK4 likely also plays a role in regulating cell division and subsequent spheroid expansion in 3D environments.

We next sought to determine if PAK4's main activity is mediated through LIMK1 (via LIMKi3). Contrary to our 2D data, LIMK1 inhibition did not significantly impact spheroid invasion nor qualitatively affect collagen densification for either SNU-475 or HepG2 spheroids. LIMKi3 also did not affect spheroid core size growth (Fig. 5b). Although we hypothesized that LIMK1 inhibition would stop 3D migration for HepG2 spheroids because it inhibited 2D cell migration, LIMK1 inhibition did not stop HepG2 cell invasion (Fig. 5a, inset 5th column) and resulted in higher percentage occupancy (Fig. 5b). We did not find any changes in collagen densification for either cell line by LIMK1 inhibition (Fig. 5d-e, Supplementary Fig. 16-17). This result suggests that inhibition of LIMK1 alone is not sufficient to reduce 3D cell migration for SNU-475 spheroids, yet for HepG2, LIMK1 inhibition is able to increase invasion.

We hypothesized that actomyosin contractility would be a key player in determining SNU-475 3D invasion due to its dominant role in inhibiting 2D invasion. Treatment with blebbistatin caused invading cells to have thinner protrusions (Fig. 5a, inset 6th column). As expected, blebbistatin treatment was able to decrease SNU-475 invasion area as well as the furthest migration distance. Co-treatment with LIMKi3 further exacerbated these effects as well as decreased percentage occupancy (Fig. 5b). Blebbistatin treatment and its co-treatment with LIMKi3 decreased short and long-range collagen densification and alignment and caused invading cells to adopt elongated shapes with thinner protrusions (Fig. 5a, inset on 6th and 7th columns, 5d). In contrast, HepG2 spheroids did not react significantly to blebbistatin treatment (Fig. 5b, Supplementary Fig. 18). These data suggest that inhibition of actomyosin contractility was able to impede SNU-475 cell migration, but does not play a significant role in HepG2 cell migration. Interestingly, ML-7 treatment caused SNU-475 invading cells to completely lose their protrusions (Fig. 5a, 8th column). In addition, ML-7 treatment caused a decrease in invasion as well as short-range and long-range collagen remodeling (Fig. 5d-e). Conversely, HepG2 spheroids did not react significantly to ML-7 treatment. Overall, these studies showed that inhibiting the cells ability to generate motile and contractile forces leads to abrogated 3D invasion for SNU-475 but not HepG2. Moreover, these studies showed how liver cancer behavior and subsequently drug response in a 3D environment can be vastly different from their responses in traditional 2D systems.

**Liver cancer stromal invasion is mediated by pathways involved in mechanotransduction.** Our 2D and 3D studies demonstrated the role of mechanotransduction in generating cell motile force as well as facilitating stromal invasion in liver cancer.

We provide a schematic that summarizes what each drug treatment will do to invasive liver cancer cells (Fig. 6a). We found that inhibition of either dynamic protrusions or actomyosin contractility is sufficient to inhibit collagen remodeling as well as subsequent liver cancer invasion. Cross-sectional images of spheroids showed distinct invasion profiles of SNU-475 cells based on different cytoskeletal perturbations (Fig. 6b). While Rac1 inhibition was able to mitigate 3D invasion, invading cells had strong actin intensity indicating the presence of tensional actin stress fibers. In contrast, cells treated with blebbistatin and blebbistatin + LIMKi3 had much thinner protrusions, which explained why these cells performed less collagen remodeling. In the extreme case, treatment with MLCK or PAK4 inhibitors caused complete loss of protrusions with invaded cells adopting a rounder phenotype. Our data support this conclusion and have been summarized in heatmap format in SNU-475 (Fig. 6b, Supplementary Figs. 19-20) cells and HepG2 cells (Fig. 6c, Supplementary Figs. 21-22). A total heatmap compares all conditions to control for both cell lines (Supplementary Fig. 23). Our data implicate PAK4, Rac1, and actomyosin contractility in the invasive properties of SNU-475 cells in 3D culture. Although LIMK1 inhibition is able to alter HepG2 2D invasion, it is unable to change HepG2 3D invasion. Overall, our in vitro tests and subsequent quantification highlight key pathways in HCC stromal invasion and identify molecular players suitable for further characterization.

**TCGA analysis reveals Rho, Rac, LIM Kinase predict poor prognosis.** We demonstrated the differential drug response between two distinctive liver cancer cell lines in both 2D cultures and a 3D collagen-based model. We then investigated whether the interrogated pathways play a role in patient outcomes by analyzing liver cancer cohorts from the TCGA database. Comparing normal liver tissue samples (225 patients) and liver hepatocellular carcinoma (LIHC) samples (371 patients), the expression of the genes: RhoA, Rac1, LIMK1, LIMK2, ROCK1, ROCK2, MLCK2, and PAK4 is upregulated in liver cancer (Fig. 7a). High expression of these cytoskeletal regulators was correlated to a poorer survival rate among 364 patients (Fig. 7b). Our clinical data analysis demonstrated a positive correlation among RhoA, Rac1, LIMK1, and MLCK2 in LIHC patients (Fig. 7c, Supplementary Fig. 24). The TCGA analysis showed that overexpression and subsequent dysregulation of this cytoskeletal machinery are associated with liver cancer. These datasets were generated from bulk RNA sequencing and, therefore, cannot differentiate between nor directly attribute these differences to tumor cells. However, these data do suggest that targeting these pathways in the liver cancer microenvironment may be a valid therapeutic strategy.

**Discussions**
Phenotypic heterogeneity poses significant challenges in cancer therapeutics. In addition to their genetic, transcriptomic, and proteomic variability, cancer cells can display a wide range of physical variability. Because evolution selects for phenotypes,

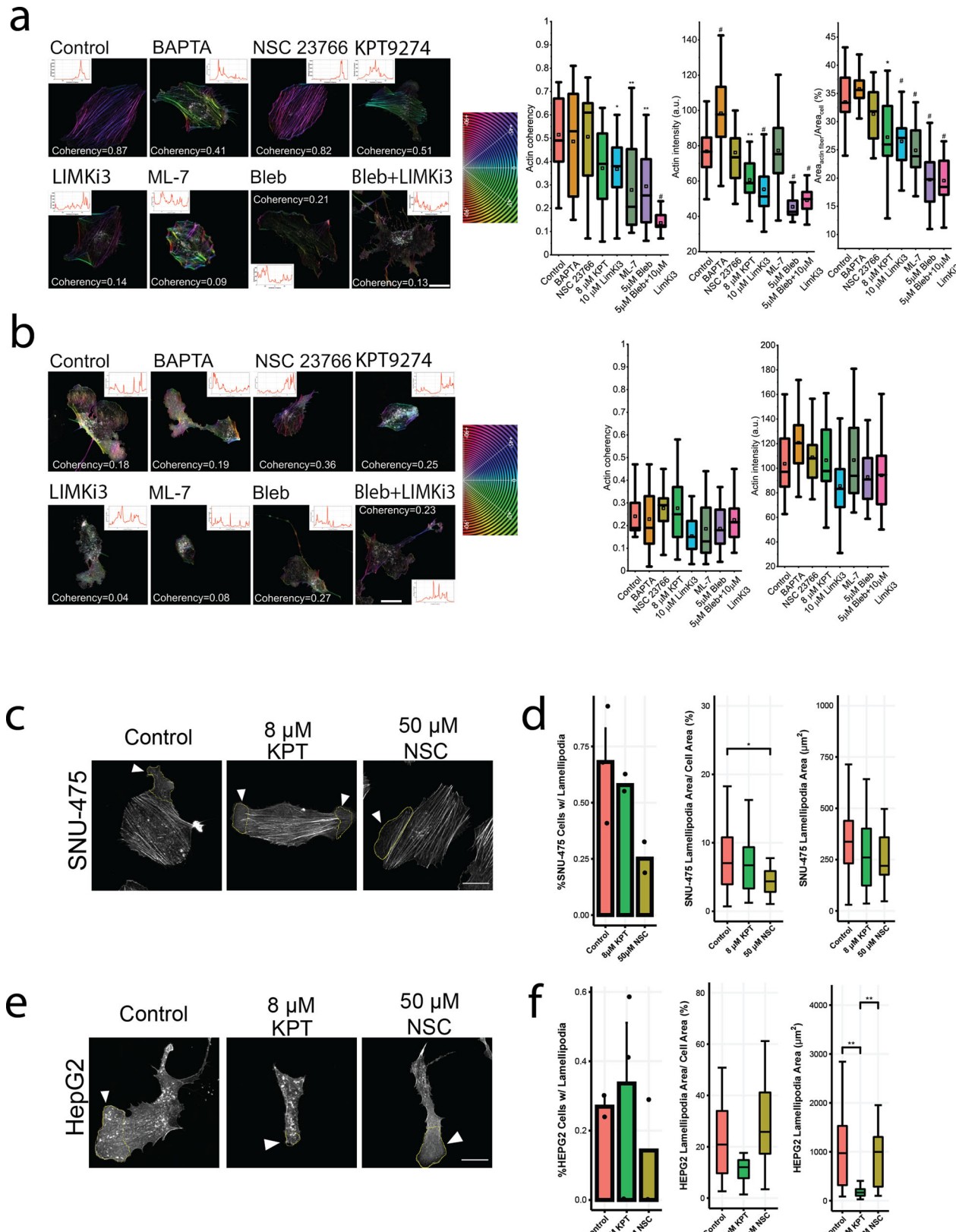

understanding the physical aspects of this heterogeneity will provide insights towards targeted drug development for different liver cancer subtypes. Our study takes an integrative approach to determine the biophysical mechanisms which underlie heterogeneous invasive behaviors and subsequent drug response in liver cancer. Based on our histological observations, we identified

diversity in liver cancer phenotypes even within the same tumor nodule. To better understand the physical implications of this diversity, we investigated the SNU-475 and HepG2 liver cancer cell lines. Whereas SNU-475 cells had distinct actin stress filaments which facilitated cell migration and resembled more stromally invasive liver cancer, HepG2 cells lacked these stress

**Fig. 4 Characterization of F-actin and lamellipodial dynamics under drug treatments for SNU-475 and HepG2.** Coherency measurement on F-actin as an indicator for cytoskeleton organization and cell morphology under drug treatments for **a** SNU-475 and **b** HepG2. Stress fiber-rich HCC cell line SNU-475 contains more organized actin and well-defined morphology and is more sensitive to the cytoskeleton drugs, compared to HepG2. Significance is compared between control and each drug condition. (From left to right) Coherency of actin stress fiber (SF) intensity and proportion of SF of total F-actin. $n > 12$ cells for each condition from $N = 2$ independent experiments. **a** Rac1 inhibitor (NSC23766) does not affect stress fiber formation but reduces lamellipodia. BAPTA seems to induce stress fiber building up and induce thin protrusions on the cell periphery. **b** The selected inhibitors do not significantly reduce total F-actin intensity for HepG2 cells. **c** Representative images denoting traced lamellipodia for SNU-475 cells. Yellow outline indicates lamellipodia tracing. **d** Metrics denoting SNU-475 number of cells with lamellipodia, lamellipodia area percentage of cell area, and lamellipodia area after no treatment, 8 μm KPT treatment, and 50 μm NSC treatment. **e** Representative images denoting traced lamellipodia for HepG2 cells. Yellow outline indicates lamellipodia tracing. **f** Metrics denoting HepG2 number of cells with lamellipodia, lamellipodia area percentage of cell area, and lamellipodia area after no treatment, 8 μm KPT treatment, and 50 μm NSC treatment. Plots show mean ± SEM. $n > 12$ cells for each condition from $N = 2$ independent experiments. One-way ANOVA with Tukey post-hoc testing was performed. Significant difference ($p < 0.05$) was detected between any two of the conditions. *$P < 0.05$, **$P < 0.01$, #$P < 0.0001$.

fibers and tended to have more globular actin and resemble more fibrotically confined liver cancer. The goal of this study was to determine how these distinct cell lines affected 2D and 3D migration.

While much of the same machinery in 2D migration is conserved in 3D migration, the presentation of this machinery in cells can vary greatly. In 2D migration, cells undergo repetitive lamellipodial protrusions, adhesion, and backside contraction[6]. In 3D migration, there are many additional factors cells face including migrating through subcellular-scaled pores, confining environments, and matrix degradation[40]. Cells in 3D are able to also generate protrusions- though typically smaller/thinner in scale compared to their 2D counterparts- into the surrounding matrix. These protrusions have a wide variety of roles: mechanically probing the environment, applying contractile forces to remodel the surrounding matrix, anchoring matrix proteases to the cell surface, and generating migratory forces[41,42]. The ability of these protrusions to extend out and generate these forces is largely regulated by actomyosin contractility and actin turnover[43]. While the loss of these protrusions via inhibition of actomyosin contractility greatly mitigates invasive properties of cells in 2D, this is not necessarily true in 3D migration. Cells have other modes of migration that are unique to 3D culture such as bleb-based amoeboid migration which typically does not require protrusions[44,45]. Actomyosin regulation in 3D has been extensively studied in other cancer types, such as breast and colorectal cancer[17,44,46]. In the context of liver cancer, while some chemical inhibitors may be able to stop invasive phenotypes on 2D surfaces, they may not be effective in stopping 3D migration. Thus, being able to differentiate between the effects of certain cytoskeletal inhibitors in distinct liver cancer types requires a more physiologically relevant 3D culture to capture the complexities of additional migration modes.

Our scratch assays showed differential response to cytoskeletal inhibition between both cell lines and demonstrated clear differences in their dissemination ability (Fig. 3b). However, the assay itself did not take into account key factors which would affect wound closure rate such as proliferation. In addition, the assay lacked the ability to capture certain modes of migration which can only be found in 3D assays. Thus, we turned to study our cell lines in 3D collagen matrices. We utilized in vitro model systems that mimic the physical tumor microenvironment and capture diverse physical responses of distinct liver cancer cell subtypes. Whereas SNU-475 cells tended to migrate in 2D and invade out of spheroids in 3D, HepG2 cells were less migratory and have more noninvasive phenotypes. In our 3D metrics, we found that all drugs tested do not significantly affect SNU-475 spheroid growth area, which is a standard metric for assessing tumor invasive capability. However, they were able to mitigate our refined readouts of functional phenotypes of invasion. Based on assays that capture distinct dynamic phenotypes that cannot

be captured by static histological and genomic studies, our study implicates cytoskeletal regulators as therapeutic targets for combinational treatments in heterogeneous liver cancers. Ultimately, these cells' response to drugs was in part due to biochemical regulation: SNU-475 cells are more mesenchymal and have a greater amount of p-MLC than do HepG2 cells which makes the SNU-475 better candidates for studying cytoskeletal regulation. One limitation of this study was the use of pharmacological agents to study these pathways. It is known that these drugs can have off-target effectors, which may also affect our results[47–49]. Future studies could address this by performing knockdowns/knockouts of members of this pathway.

Lamellipodia-mediated migration is a key mode of metastasis in cancer, and Rac1 is a key regulator of lamellipodia and was found to be upregulated in our TCGA analysis[50]. We found that inhibition of Rac1 significantly inhibits SNU-475 invasiveness but did not significantly affect such metrics for HepG2 cells. Although HepG2 cells tended to have larger lamellipodia-like structures than SNU-475 cells do, this did not correlate with increased invasion. Interestingly, our immunofluorescence results revealed that Rac1 inhibition will significantly decrease lamellipodia size in SNU-475 cells but not in HepG2 cells (Fig. 2a-b). Previous studies have also implicated the importance of Rac1 in mediating long-range collagen alignment and densification[17,51] and, therefore, may help facilitate invasive behavior. This was consistent with our 3D studies as Rac1 inhibition caused less long-range collagen density (Fig. 5d). Thus, understanding the role of Rac1 and its interplay among cytoskeletal proteins as well as its role in cell-ECM interactions will be crucial in understanding liver cancer cell invasiveness.

It has been previously reported that matrix remodeling and degradation are necessary for cancer stromal invasion[52]. This remodeling is in large part due to dynamic protrusions and actomyosin contractility which help the cell adhere to surrounding collagen and apply tensile force[31]. We note that inhibition of either of these two pathways—via Rac1 or actomyosin inhibition—were sufficient to significantly decrease collagen remodeling and subsequent stromal invasion in SNU-475 cells. Our bioinformatic studies further suggested the importance of these components as their molecular mediators are significantly upregulated in patients with LIHC. While our study focused on the mechanical basis of collagen remodeling in 3D matrices, chemical signals such as MMPs which degrade collagen are also necessary for remodeling and invasion to occur[53]. Future studies will examine how altering degradation capabilities will affect differential invasive phenotypes in liver cancer.

We next looked at PAK4 inhibition, which is upstream of both actin turnover processes and actomyosin contractility. We found that PAK4 is able to significantly decrease wound closure rate, decrease the presence of actin stress fibers and reduce actin fiber thickness, and decrease 3D spheroid invasiveness in both cell

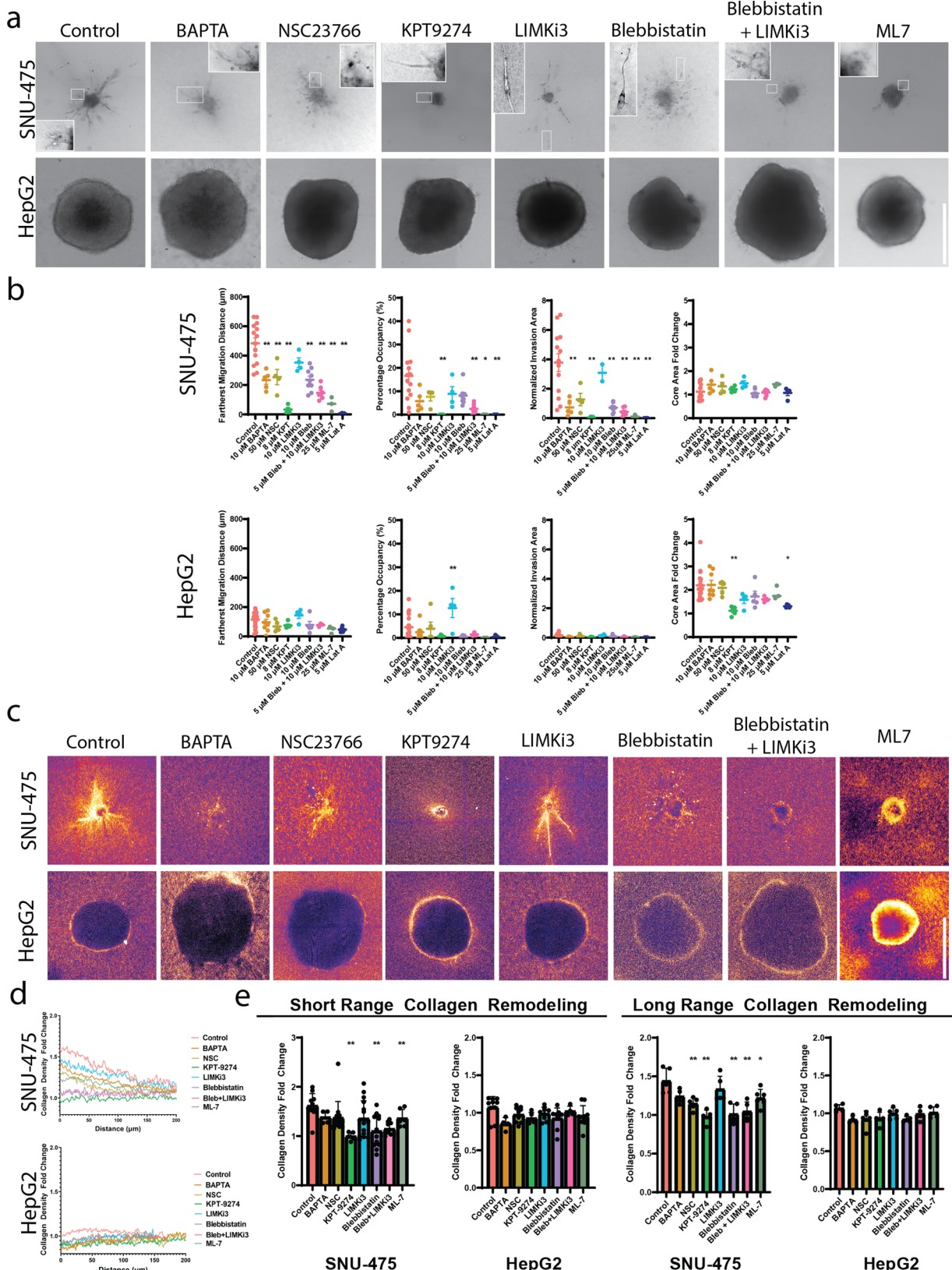

lines. Interestingly, PAK4-inhibited SNU-475 cells were still able to form stress fibers and lamellipodia in 2D. Notably, PAK4 inhibition in HepG2 drastically reduced cell size and disseminated cell area. We hypothesized that these marked decreases in invasive behavior may be due to decreases in cofilin-mediated actin turnover. One way to modulate this is through inhibition of

LIMK1 which would create more free actin barbed ends and inhibit protrusions[54]. We found that inhibiting cofilin phosphorylation via LIMK1 inhibition was able to stifle 2D cell migration and reduce actin fiber thickness but was not able to decrease 3D spheroid invasiveness in both cell lines. Similarly, it has previously been reported that inhibition of LIMK1 weakens

**Fig. 5 Characterization of SNU-475 and HepG2 3D spheroid invasion under drug treatments. a** Representative images of SNU-475 (top row) or HepG2 (bottom row) spheroids 5 days after seeding in collagen gels. Arrows indicate circular morphology of invading SNU-475 cells. Scale bar, 500 μm. **b** We use fractional occupancy, normalized invasion area, and farthest migration distance to quantify the spheroid invasiveness, and calculate the core area fold change on day 5 normalized by that of day 0 to quantify the spheroid size growth for SNU-475 and HepG2 spheroids. $n > 4$ spheroids for each condition from $N = 2$ independent experiments. **c** Representative images of reflectance microscopy to indicate collagen densification and alignment for NSC23766, KPT9274, Blebbistatin, and Blebbistatin + LIMKi3 treatment for both cell lines. Scale bar, 500 μm. **d** Average normalized collagen density profiles for SNU-475 SNU-475 (top row) or HepG2 (bottom row) spheroids 5 days after seeding in collagen gels. Measurements are normalized to the last 30 data points of each plot profile. Plots show mean collagen density. **e** Quantification of short-range and long-range collagen remodeling. The short-range collagen remodeling metric is determined as the collagen density fold change immediately outside the periphery of the spheroid (30 μm). Long-range collagen density is determined as the collagen density fold change 80 μm away from the spheroid boundary. Plots show mean ± SD. $n > 4$ spheroids for each condition from $N = 2$ independent experiments. One-way ANOVA with Tukey post-hoc testing was performed. Significant difference ($p < 0.05$) was detected between any two of the conditions. *$P < 0.05$, **$P < 0.01$.

---

actin stability and collective cell migration but does not significantly affect single-cell invasion[55]. This may explain why our results do not show the decreased invasion of SNU-475 cells after LIMK1 inhibition.

We then investigated the effects of inhibiting actomyosin contractility. Although we were able to see decreases in 3D SNU-475 spheroid invasiveness after treatment with blebbistatin, cells were still able to form thin protrusions. These thin protrusions were not able to induce significant collagen remodeling. This suggested that cells were able to probe their surrounding environment but were not tensile enough to produce contractile and migratory force. Cells treated with MLCK inhibitor, which also blocks actomyosin contractility, showed decreased 3D spheroid invasion in SNU-475s. However, cells were more rounded with less prominent protrusions. Overall, this demonstrated that 3D cultures are able to distinguish between these distinct phenotypes arising from differential regulation of actomyosin contractility. These differences can arise from the regulation of other components of this pathway such as ROCK and RhoA. Future experiments can help decouple the role of these components in 3D migration.

Immunofluorescence in both 2D and 3D results revealed that PAK4-inhibited cells look markedly different from cells treated with LIMKi3, ML-7, or blebbistatin (Figs. 3c, 6b). This may suggest that this combination did not fully inhibit cell contractility and actin stress fiber formation. Another possibility is that PAK4 may mediate cell migration in liver cancer through other forms of regulation. PAK4 forms a protein complex with NAMPT, which is crucial for mitochondrial function and has been implicated in cancer metabolism and stemness[56]. Future studies targeting these pathways will further delineate the role of PAK4 in liver cancer cell invasiveness.

Our bioinformatic analyses suggested the correlation of some of the markers examined in this study to HCC. This was in alignment with previous bioinformatic studies which have shown the correlation of cytoskeletal markers, such as LIMK1, to a higher hazard ratio of death in liver cancer patients[57]. However, these data were generated from bulk RNA sequencing and cannot directly attribute these differences to malignant components of the tumor. For example, these differences could also be attributed to hepatic stellate cells, which are the main effectors of liver cirrhosis and may be linked to liver cancer[58]. Future studies can use IHC to determine if stromally invasive liver cancer cells in patients have cytoskeletal dysregulation. Furthermore, these studies should include staging information of these datasets to correlate this dysregulation with liver cancer progression.

Liver cancer is a complex disease composed of a heterogeneous mixture of cancer cell subtypes, and therapeutic strategies can benefit by targeting each subtype. Here we present clinical and in vitro data that highlight liver cancer subtype-specific responses to cytoskeletal perturbations. We find that cytoskeletal perturbations more drastically inhibit the more migratory SNU-475 cells but less apparently on the more proliferative HepG2 cells. Further, signaling programs associated with motility are correlated with a worse prognosis. By analyzing functional outputs of invasion, we develop key metrics of cancer aggressiveness that capture the dynamic behaviors of our cell lines and their responses to cytoskeletal perturbation. Our bioinformatic analysis correlates cytoskeletal dysregulation with liver cancer prognosis. Our integrative approach highlights phenotypic variations driven by cytoskeletal signaling, their consequences and coexistence within the same patient tumor, and how to evaluate drug efficacy in targeting critical biophysical phenotypes that may be masked in traditional drug screens against tumor growth.

## Methods

**Histology case selection.** HCC cases were selected from the Yale New Haven Hospital pathology database. The pathology slides were evaluated for background liver disease and the presence or absence of cirrhosis. All available hematoxylin and eosin slides were reviewed for each case by 2 pathologists (X.Z. and M.E.R.), and a consensus was reached. The images of the pathology slides were acquired with an EVOS imaging system (Invitrogen).

**Cell culturing.** SNU-475 and HepG2 cells were obtained from the American Type Culture Collection (ATCC; Manassas, VA, USA). ATCC validated all cell lines by Short Tandem Repeat Analysis. Cell lines were maintained at 37 °C, 5% $CO_2$. SNU-475 cells were maintained in Roswell Park Memorial Institute (RPMI) medium supplemented with 10% fetal bovine serum, 1% L-glutamine, and 1% penicillin/streptomycin. HepG2 cells were maintained in Dulbecco's Modified Eagle's Medium (DMEM) medium supplemented with 10% fetal bovine serum, 1% L-glutamine, and 1% penicillin/streptomycin. Media was changed every other day, and cells were split every 3-4 days.

**Spheroid preparation.** Spheroids were composed of either SNU-475 or HepG2 cells. They were prepared using established protocols[59]. The boiled 20% agarose + 1× Phosphate Buffered Saline (PBS) mixture was quickly pipetted 50 μl to each well of a 96-well plate and allowed to gel. We then trypsinized our cells and aliquoted a subset of these cells to fresh media and adjusted the volume to achieve a final volume of 10 cells/μl. We then pipetted 100 μl of the cell solution into each well of the 96-well plates, resulting in spheroids made up of 1000 cells. To promote the spheroid formation, we centrifuged our cells at 400 relative centrifugal force (rcf) for 10 min. Spheroids were incubated at 37 °C for 4 days.

On the fourth day, we pretreated our resulting spheroids with drugs of interest for 2 h at 37 °C prior to seeding them in the collagen gel. The spheroids were then picked up with a wide-tip p1000 pipette tip and transferred to a 1 ml Eppendorf tube. Spheroids were then incubated with 10 μM CellTracker GFP for 20 min. Without washing at this stage, these spheroids were immediately transferred into a collagen solution. This spheroid-containing solution was then pipetted into glass-bottom 24 wells with an average of 2–3 spheroids per gel and a final volume of 50 μl. To ensure that spheroids were embedded in the middle of the gel, we flipped our plates in intervals of 1 min, 1 min and 30 seconds, and 3 min at 37 °C. The plate was then allowed to polymerize at 37 °C for 1 h. After the 1 h incubation, three PBS washes were performed to remove residual CellTracker.

**Confocal microscopy.** A Leica SP8 laser scanning confocal microscope with a ×10 objective (Wetzlar, Germany) was used to image spheroids. A temperature of 37 °C and a 5% $CO_2$ atmosphere were maintained using a humidified OKO labs live-cell imaging incubator. For 2D immunofluorescence experiments, z-stacks of all cells were imaged. Cell outlines were manually traced to obtain metrics for cell shape, and automatic Jython (ImageJ Python) scripts were used to trace the nuclear shape.

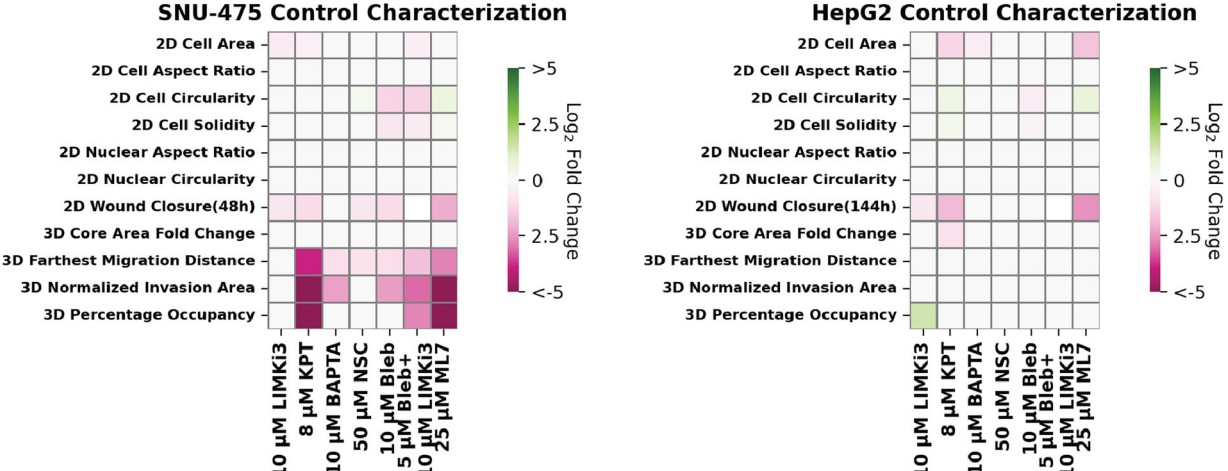

Custom scripts were developed to measure cell and nuclear aspect ratio, circularity, area, and aspect ratio. For 3D collagen spheroids, z-stacks of all spheroids were imaged. Z-stack snapshots were taken on 0, 1, 3, and 5 days.

**Embedding SNU-475 and HepG2 spheroids in collagen gels**. A collagen gel was made by adding sufficient 0.5 N NaOH to neutralize a mixture of double-distilled H$_2$O, 10× PBS, and acetic-acid-solubilized type I rat tail collagen (Corning, Corning, NY, USA) on ice for a final collagen concentration of 2 mg/ml. In order to track collagen, we fluorescently label collagen with carboxylated polystyrene Alexa-Fluor 660/680 beads (ThermoFisher Scientific, cat.# F8807). Prior to depositing the collagen, plates used for collagen gel seeding were surface-coated with poly-dopamine, as previously described[60,61]. This coating allows collagen gels to stick to the surface of the plates and prevents detachment of the gel. Drug-primed

**Fig. 6 HCC invasion schematic and heatmap summary. a** Schematic illustrating how altering cytoskeletal dynamics will affect collagen densification and subsequent migration. **b** Representative immunofluorescence images of SNU-475 spheroids (first row) and HepG2 spheroids (second row) treated with BAPTA (Ca2+ chelator), NSC23766 (Rac1 inhibitor), KPT9274 (PAK4 inhibitor), LIMKi3 (LIMK inhibitor), blebbistatin (myosin inhibitor inhibitor), both blebbistatin and LIMKi3, or ML-7 (MLCK inhibitor), respectively. Green represents F-actin and magenta is DAPI. Scale bar: 50 μm. Heatmap illustrating 2D and 3D metrics characterized in this study for **c** SNU-475 and **d** HepG2 cells. Heatmap values are colored if the difference between two conditions is considered significantly different ($p < 0.05$). White indicates no significant difference between conditions. Color intensity is determined by log2 fold change of the group on the x-axis over control. Green signifies upregulation while pink signifies downregulation. One-way ANOVA with Tukey post-hoc testing was performed to test for significance.

---

spheroids incubated with CellTracker GFP were then added to the gel and transferred to a 24-well glass-bottomed cell culture plate (MatTek, Ashland, MA, USA) kept on an ice pack. Once all gels were transferred to the 24-well plate, the plate was transferred to an incubator at 37 °C with 5% $CO_2$. The sample was flipped several times during gelation to prevent spheroid sedimentation at the bottom of the plate. After 1 h, three PBS washes were performed to remove residual Cell-Tracker GFP. Finally, fresh media with appropriate drugs were added to each well and were subsequently maintained at 37 °C with 5% $CO_2$. Drug media was replaced every other day.

**2D collagen coating**. 2D collagen coating was performed by dissolving acetic-acid–solubilized type I rat tail collagen (Corning, Corning, NY, USA) in 0.02 M acetic acid to a final concentration of 50 μg/ml. This solution was then pipetted onto a 15-well glass plate (Ibidi) and allowed to incubate at 37 °C for 1 h. Each well was then washed with 1× PBS and allowed to dry. Cells were immediately plated.

**2D Immunofluorescence**. Fifteen-well glass plates (Ibidi) having a growth area of 0.125 cm$^2$ were collagen-coated as described in the previous section. Three hundred SNU-475 or HepG2 cells were plated and allowed to attach for 24 h before the addition of drug media. After 24 h of incubation, the media was removed, and each well was washed three times briefly with 1× PBS. Fixation was performed for 1 h at room temperature with a 4% paraformaldehyde solution. Following three washes with PBS, cells were permeabilized with 0.1% Triton X-100 for 10 minutes. After two washes with PBS, the cells were then blocked with a 1% bovine serum albumin (BSA) solution for 1 h. Following blocking, cells were incubated with a rabbit anti-NCS1 primary antibody diluted in blocking solution (sc-13037, diluted 1:100; Santa Cruz Biotechnology) overnight at 4 °C. After extensive washing with PBS, cells were incubated with an Alexa Fluor-488 goat anti-rabbit secondary antibody (diluted 1:1000; Thermo Fisher Scientific) for 2 h at room temperature in the dark. After extensive washing with PBS, cells were then incubated with an Alexa Fluor-647-phalloidin (1:500 dilution) (Thermo-Fisher Scientific) and Hoechst 33342 (1:2000 dilution) (ThermoFisher Scientific) solution overnight. The samples were then extensively washed.

**3D immunofluorescence**. After 5 days of culture in collagen gels, spheroids were fixed with 4% paraformaldehyde solution at 37 °C for 1 h. Following thorough washes with PBS, fixed samples were embedded in Tissue Tek OCT solution and frozen in 2-methylbutane surrounded by liquid nitrogen. Samples were cut in 50 μm slices and placed on poly-lysine coated slides. Samples were then washed three times with PBS and then incubated with 0.1% Triton X-100 solution for 20 min. Samples were then washed briefly three times before being incubated with 1% BSA solution for 1 h. Following blocking, cells were incubated with mouse Ki67 (1:100; Santa Cruz; sc-23900) overnight at 4 °C. After extensive washing with PBS, cells were incubated with an Alexa Fluor-647 goat anti-mouse secondary antibody (diluted 1:1000; Thermo Fisher Scientific, A-21235) for 2 h at room temperature in the dark. After extensive washing with PBS, cells were then incubated with an Alexa Fluor-555-phalloidin (1:500 dilution) (ThermoFisher Scientific) and Hoechst 33342 (1:2000 dilution) (ThermoFisher Scientific) solution overnight. Samples were mounted with Vectashield Antifade mounting medium and sealed. Samples were stored at 4 °C before imaging.

**Image analysis of 3D spheroid invasion assay**. To quantify the invasiveness of the two cell lines, the images of the cellular spheroids that have been cultured in collagen gel for 5 days were analyzed. First, we manually traced the disseminated cells and also the spheroid core, respectively. Similar to what has previously been used as a spheroid core in image analyses of spheroid invasion assay[62], the spheroid core in this report was defined as the dark dense region of the spheroid seen in brightfield images, and the disseminated areas were any areas occupied by cells outside of the dense core (Supplementary Fig. 25). We calculated the distribution of these disseminated cells by calculating the distance of each pixel in the disseminated cell area to the spheroid core-periphery. To assess the invasiveness, we calculated these metrics: percentage occupancy, normalized invasion area, and farthest migration distance. To calculate the percentage occupancy, we divided the area occupied by disseminated cells at a certain distance by the total ring area at that distance. To compare between the cell lines and across drug treatments, we compared the percentage occupancy at a characteristic distance relative to the control group. The characteristic distance is 100 μm for SNU-475 and 50 μm for HepG2. To calculate the normalized invasion area, we divided the

total area of the disseminated cells on day 5 by that spheroid's size on day 0. The normalization is to correct for spheroid size. To calculate the farthest migration distance, we calculated the farthest distance travelled by any disseminated cells of a single spheroid and used that value to represent the invasiveness of that spheroid. Algorithms were implemented in custom MATLAB codes. Statistics analysis was performed using GraphPad Prism.

To quantify the spheroid core growth, we manually traced and quantified the spheroid core areas on 0, 1, 3, and 5 days. In the normalization step, we divided the spheroid area of each spheroid on day 5 by its area on day 0. Both manual tracing and quantification were implemented in MATLAB. Statistics analysis was performed using GraphPad Prism.

**Collagen density calculation**. To quantify the bulk collagen remodeling by spheroids to their surrounding environment, we calculated the collagen density fold change as a function of distance from the spheroid. Each day-5 spheroid was first manually outlined in the brightfield channel, and then an average Z projection was taken in the reflectance channel. Using the trace and reflectance image, we calculated the centroid and drew radial line segments that were angled π/6 rads or 15° from each other. The line segments were long enough to reach a distant region with collagen intensity comparable to the background. For each radial line, we took plot profiles distanced 30 μm from the edge of the spheroid and then averaged the plot profiles to obtain average intensity versus distance. In the averaged profile, we took the maximum intensity to be the first point of each profile and the plateau intensity to be the average of the last 30 points of the profile. We then divided every value by the plateau intensity to get collagen density fold change. The short-range collagen density fold change is defined as the collagen density fold change 30 μm away from the spheroid (the first measurement of the average collagen density fold change). The long-range intensity fold change is defined as the collagen density fold change 80 μm away from the edge. We note here that we used the confocal reflectance signal as a label-free relative measure of collagen density, as collagen has a high reflectance signal, but reflectance is not specific for collagen alone. The original outline of the spheroid was manually traced in ImageJ while the rest of the calculations were performed in Python 3.

**Scratch assays**. Thirty thousands of SNU-475 cells or 37,500 HepG2 cells were plated in sterile collagen-coated plastic 24-well plates. The cells were allowed to attach and grow for 24 h before the removal of media and subsequent addition of 1X PBS. A scratch was then made with a p1000 pipette tip in each well. The PBS was removed from each well and replaced with appropriate media with drugs. Each well was immediately imaged to obtain 0 h time points. For SNU-475 cells, images were taken at 0, 12, 24, 36, and 48 h; the wound had closed by 48 hours. For slower-moving HepG2 cells, images were taken at 0, 12, 24, 36, 48, 72, 96, 120, 144 h; the wound had closed by 144 h. Media with appropriate drug concentrations was changed every other day.

**Western blots**
*Cell treatment*. Cells were treated in six-well plates with 2 ml of cell media (see Cell Culturing) for 48 h with their corresponding drugs. Prior to cell treatment, a hemocytometer was used to count cells treated with trypan blue to ensure 100,000 to 200,000 SNU-475 or 700,000 to 1,000,000 HepG2 cells per well. Following the 48 h treatment, cells were immediately lysed (see below). The drugs tested and their concentrations are listed in Table 1.

*Cell lysis, sample preparation, and electrophoresis*. A lysis buffer consisting of protease inhibitor and mammalian protein extraction reagent (MPER) was used to lyse cultured cells. The resulting solution was spun at 13,000 revolutions per minute (rpm) for 20 minutes at 4 °C, and the supernatant was preserved for the western blot. A Bicinchoninic acid assay was used to measure protein concentration. Western blot samples were prepared by mixing the cell culture protein, loading buffer, and reducing agent. Equal amounts of proteins were added for all samples run on the same gel, and water was used to attain equal volumes. Protein samples were boiled at 95 °C for 5 min and then loaded to 4–12% NuPAGE gradient Bis-Tris polyacrylamide protein gels for electrophoresis in 2-(N-morpholino) ethanesulfonic acid (MES) buffer.

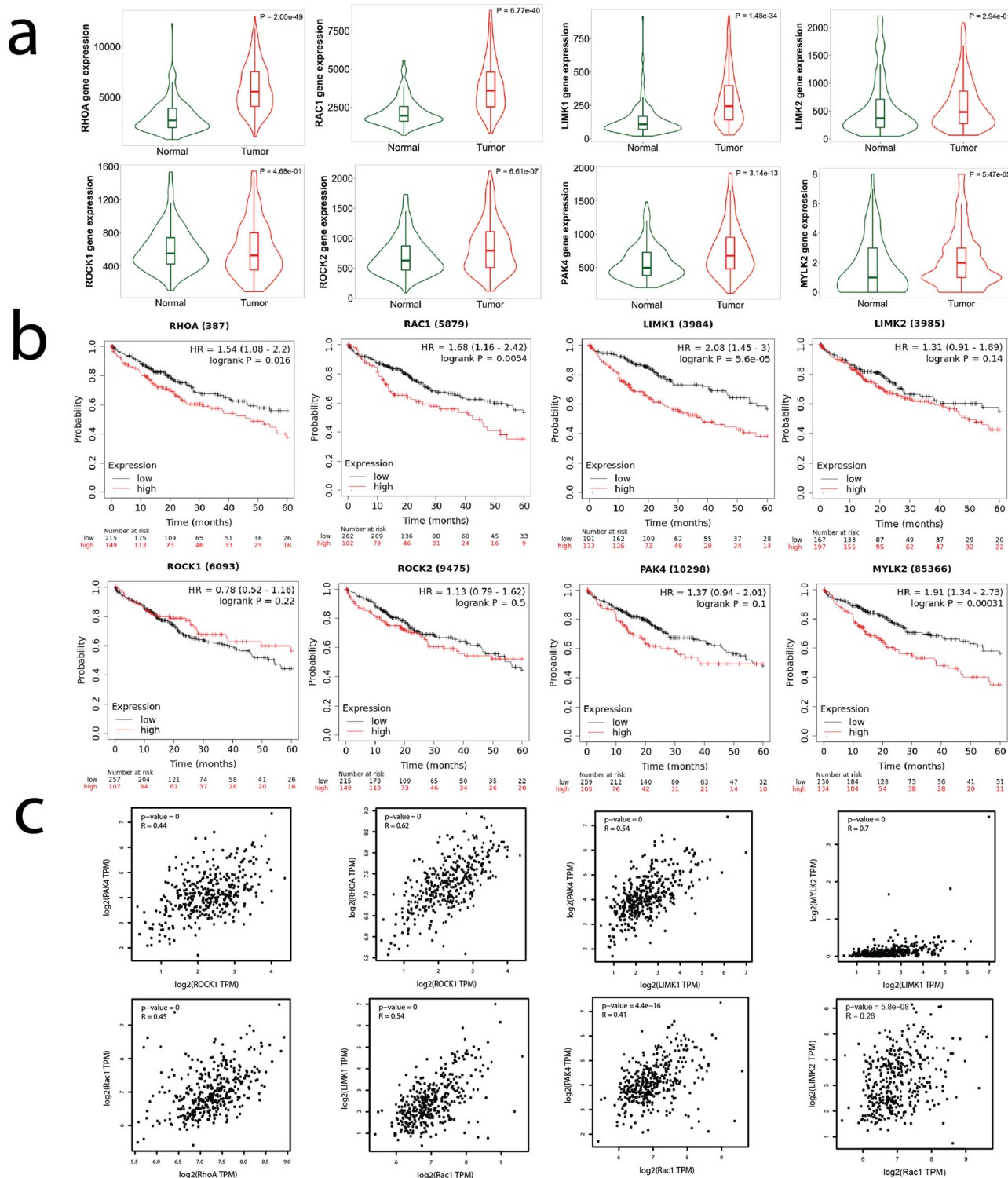

**Fig. 7 Rho family of small GTPases, PAK4, LIM kinase, and myosin light chain kinase 2 (MYLK2) are correlated with more invasive liver cancer.**
**a** Gene expression levels of RhoA, Rac1, LIMK1, LIMK2, ROCK1, ROCK2, PAK4, and MYLK2 are upregulated in HCC compared to normal liver tissue.
**b** Kaplan–Meier analysis ($n = 364$ patients) reveals that RhoA, Rac1, LIMK1, LIMK2, ROCK1, ROCK2, PAK4, and MYLK2 are linked to poor prognosis of HCC. **c** Correlation analysis shows expressions of Rac1, RhoA, and LIMK1 are positively correlated, suggesting the signaling axes Rac1-LIMK and RhoA-LIMK may be active in HCC. For each correlation plot, $p < 10^{-16}$.

*Chemiluminescent western blot detection.* A wet transfer was used to transfer protein from the gels to a polyvinylidene fluoride (PVDF) membrane at 100 volts for 2 h in a transfer buffer. The membrane was then blocked in 5% milk in Tris-buffered saline and Polysorbate 20 (TBST) for an hour at room temperature. The membrane was incubated overnight at 4 °C with its respective primary antibody. The membrane was then washed for 5 min three times in TBST and incubated with its corresponding secondary antibody for two hours at room temperature. The blot was washed again and then processed using a ECL West Dura kit and visualized using autoradiography film using different exposure times. The resulting film was quantified using ImageJ and statistical analysis was performed using GraphPad Prism.

**Table 1 Drug treatment dosages.**

| Condition | Concentrations (μM) |
|---|---|
| BAPTA | 3, 10 |
| Sorafenib | 3, 10 |
| Chelethyrine chloride | 3, 10 |
| ML141 | 3, 10 |
| NSC23766 | 15, 50 |
| KPT9274 | 3, 10 |
| LIMKi3 | 3, 10 |
| (S)-4'-nitro Blebbistatin | 5, 10 |
| (S)-4'-nitro Blebbistatin + LIMKi3 | 5 + 5 or 10 |
| Lithium | 300, 1000 |

**Table 2 Antibody dilutions.**

| Antibody | Part number | Host | Dilution |
|---|---|---|---|
| IP3R1- 1° | Homemade YU272 | RABBIT | 1:1000 |
| PAK4- 1° | Cell signaling 3242 S | RABBIT | 1:1000 |
| B-ACTIN- 1° | Cell signaling 13E5 (#4970 L) | RABBIT | 1:5000 |
| B-ACTIN- 1° | Cell signaling 8H10D10 | MOUSE | 1:5000 |
| NCS1- 1° | ab129166 | RABBIT | 1:1000 |
| P-COFILIN- 1° | Cell signaling S3 77G2 (3313 S) | RABBIT | 1:1000 |
| T-COFILIN- 1° | Cell signaling 8503 S | RABBIT | 1:1000 |
| P-MLC- 1° | ab2480 | RABBIT | 1:500 |
| IRDye® 800CW Anti-rabbit | LI-COR 926-32213 | Donkey | 1:30,000 |
| HRP Anti-Rabbit | BIO-RAD #1662408EDU | Goat | 1:50,000 |
| HRP Anti-Mouse | BIO-RAD #1706516 | Goat | 1:50,000 |

*Near-infrared western blot detection.* A wet transfer was used to transfer protein to a Nitrocellulose membrane at 100 volts for two hours in the transfer buffer. The membrane was then blocked in 1% TBS Casein Blocker for an hour at room temperature. The membrane was then incubated overnight at 4 °C with its respective primary antibody. The membrane was then washed for 10 min three times in TBST and incubated with its corresponding secondary antibody for one hour at room temperature. The blot was washed again and then processed using an Odyssey Imaging System. Image Studio was used for quantification. Antibodies used can be found in Table 2.

*Bioinformatics analysis.* Using a recently published online bioinformatic tool TNMplot[63], we compared gene expression of RhoA, Rac1, LIMK1, and MYLK2 between HCC ($n = 371$) from the TCGA project[64] and normal liver tissue ($n = 225$) pooled from both TCGA and Genotype-Tissue Expression (GTEx) portals[65]. Mann-Whitney U test was used for statistical significance. Sixty-month survival analysis as a function of the genes of interest from 364 HCC patients from the TCGA was conducted using a published online Kaplan–Meier Plotter[66]. The cutoff value used to determine "High" and "Low" groups in the survival analysis was selected as the best performing value among all possible cutoff values. Gene expression correlations were generated by the bioinformatic tool GEPIA2 based on the TCGA-LIHC database[67].

*Statistics and reproducibility.* GraphPad Prism and RStudio were used for all statistical analyses other than for the TCGA data. Specific methods of statistics, *P*-values, and sample numbers of each comparison are reported in figure legends. For 2D immunofluorescence staining, the sample size is 2 biological replicates with 18 cells from each replicate. For scratch assays, the sample size is 2–3 biological replicates with at least 3 independent wounds per condition. For 3D spheroid experiments, the sample size is 2–3 biological replicates with at least 4 spheroids per condition. For Western blots, we did 3 biological replicates for each condition. Error bars for bar plots show mean ± SEM.

## Data availability

All data supporting the results in this study are available within the Article and its Supplementary Information. Manuscript raw data including raw western blots and data points can be found at (https://figshare.com/articles/dataset/FigShareFile_xlsx/17999852). All other data are available from the corresponding author upon reasonable request.

## Code availability

Algorithms were implemented in custom Python3 and MATLAB codes and are available from the corresponding author upon reasonable request.

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

## Acknowledgements

R.Y.N. was supported by NIH grant T32EB019941. We acknowledge support from the National Institutes of Health National Institute of General Medical Sciences grant number R35GM142875 to M.M. and National Institute of Diabetes and Digestive and Kidney Diseases grant 5P01DK057751 to B.E.E. This project was supported in part by a Yale Liver Center Pilot Project under award NIH P30 DK034989. T.T.F. was supported by a scholarship from the German Academic Scholarship Foundation. We also thank Professor Rong Fan and Dr. Yanxiang Deng for the access to the EVOS imaging system and their assistance.

## Author contributions

R.Y.N., H.X., X.G., A.A., X.Z., M.E.R, B.E.E., and M.M. designed the study. R.Y.N., X.G., and K.M.F. conducted the 2D and 3D immunofluorescence experiments and 3D spheroid experiments. R.Y.N., H.X., X.G., A.T.C., and M.M. developed methods for analyzing data and analyzed the microscopy data. A.A., T.T.F., and R.Y.N. performed the western blots. X.G. and X.Z. performed microscopy on all histology slides. X.Z. and M.E.R. provided all pathology assessments. All authors contributed to writing or editing the manuscript.

## Competing interests

The authors declare the following competing interests: B.E.E. is a cofounder of Osmol Therapeutics, a company that is targeting NCS1 for therapeutic purposes. The remaining authors declare no competing interests.
