## [Peer Review File · Communications Biology]

Reviewers' comments:

Reviewer #1 (Remarks to the Author):

The authors sought to take an integrative approach to explore the role of cytoskeletal regulation on the phenotypic diversity in liver cancer. They compare two cell lines (one mesenchymal-like, one hepatoblast-like) using a multitude of cytoskeletal inhibitors, and a variety of functional and biophysical readouts in both 2D and 3D contexts. They then attempt to relate this information to human liver tumor samples as clinical relevance. The studies are extremely well done with proper controls and statistical rigor within each cell line. However, as only 2 cell lines are analyzed (and each is from a different phenotypic class), it is difficult to determine the broader relevance to human biology. This is particularly true as HepG2 cells lack almost any migratory capability in 3D, so the usefulness of this cell line is unclear in terms of experiments aimed at determining response to cytoskeletal inhibition. While I agree that they authors may have identified key pathways in HCC stromal invasion, these pathways have been described in other cancers, and the current study focused on 1 invasive cell line lacks sufficient rigor for such a broad claim.

Major concerns:

While the authors show that 2 types of cells exist in liver tumors *in vivo*, it is unclear if these are related to the cell lines chosen for analysis by anything other than histology. Molecular analysis of these cell types *in vivo* (such as IF of specific markers also tested in the 2D cell lines) would allow a more direct comparison.

The TCGA data clearly show a relationship between the chosen cytoskeletal regulators and presence or absence of liver cancer. However, this does not shed any light on the relationship between these regulators and invasive capabilities of human liver tumors. Without this information it is unclear how this data is relevant to the 2D/3D experiments performed throughout the manuscript. I disagree that these data "implicates cytoskeletal dysregulation as a marker of liver cancer progression" since different stages of liver cancer are not analyzed.

Minor points:

Fig 2B - it is unclear why the bottom panels have 4 lanes across and why the samples were run on separate gels, making a direct comparison difficult.

The text refers to Fig 5D, but this doesn't appear in the actual Figure.

Reviewer #2 (Remarks to the Author):

This study aims at understanding responses of phenotypically distinct liver cancer cell lines to different chemical compounds targeting major signaling cascades that remodel the actin cytoskeleton. The authors used both 2-D and 3-D systems to investigate cancer cell migration, morphology and actin filaments assembly and tested variety of compounds that inhibit actin filament turnover and actomyosin contractility. Their experimental system included a highly metastatic SNU-475 and a poorly invasive HepG2 cell lines. The authors report differential effects of the cytoskeletal drugs on invasive versus non-invasive liver cancer cell lines and also their differential responses in the 2-D versus 3-D systems. Overall, the study has scientific merit, since the details of cancer cell responses to actin cytoskeleton targeting compound are poorly understood and such compounds could be used to develop new antimetastatic therapies. Furthermore, the described experiments are well-performed, their results are robust and carefully interpreted. Nevertheless, the present manuscript has several issues with experimental design and the presented data.

Comments:

1. The choice of actin-cytoskeleton targeting drugs is poorly justified. A significant number of different chemical compounds is known to modulate actin filament turnover and actomyosin contractility. There is no rational explanation/ justification for the selection of specific chemicals in the present study. It seems that the study largely probes different steps of the Rac1 small GTPase signaling and it would be better if the author reshape their Introduction to get it more focused on

Rac1 and its roles in liver cancer dissemination. Likewise, if the focus of this paper on a broad Rho family of small GTPases, this should be better described.

2. Along the same lines, using Sorafenib or BAPTA does not fit well into the design of this study. It seems that Sorafenib was used as a positive control, as a known inhibitor of cancer cell motility, which does not add much to the study. Ca chelator, BAPTA, would be a reasonable choice if the authors investigate calcium sensitive mechanisms of actomyosin contractility that involve Ca- and calmodulin-dependent myosin light chain kinase (MLCK). In this scenario, they should add MLCK inhibitors to their panel.

3. While using chemical inhibitors of the actin cytoskeleton is an acceptable experimental approach to probe cytoskeletal mechanisms of cell phenotype and functions, the authors should discuss study limitations, which is possible nonspecific off-target effects of these compounds. This is particularly relevant to the Rac inhibitor NCS23766. Also, the authors should report if the inhibitors affect cell viability and cell proliferation, especially under conditions of long exposure (up to 5 days), which could affect interpretation of their results.

4. Immunoblotting data in Figure 2B and 3E lack important controls, such expression of total MLC and cofilin.

5. Extracellular matrix remodeling was used as one of functional readouts, but the presented reflectance microscopy images in Fig 2E and 5C should be quantified.

6. The choice of targets for the correlation analysis presented in Figure 7 is not optimal. MYLK2 does not really fit into Rho-GTPases pathways and was not investigated in the experimental part of this paper. Why only LIMK1 isoform was included? What about LIMK2 expression? Since the authors examined RhoA expression it would be logical to include immediate downstream effectors of RhoA, such as ROCK1 and ROCK2. Why PAK4 was omitted from the analysis since PAK4 inhibition was extensively probed in the experimental part of the paper.

7. The note that actin targeting actin filament turnover and actomyosin contractility may have different effects on cell migration and invasion under 2-D and 3-D conditions is not novel and has been previously described for other cancer types (see for example PMC2630553 and PMC7916823). This should be properly discussed and cited.

8. Discussion is shallow and largely repeats the Result section. The authors should put more in depth discussion of how coordinated actin filament turnover and actomyosin contractility drives liver cancer cell invasion and how these mechanisms could be different under 2-D and 3-D environment.

Reviewer #3 (Remarks to the Author):

Nguyen et al. describe two distinct liver cancer cell behaviours observed in clinical samples, stroma-invasive and non-invasive. To understand these behaviours, the authors use two cell lines, a collagen-invasive with a mesenchymal phenotype and a collagen non-invasive with a more epithelial phenotype. Through inhibition of protein activity, the authors observe distinct responses in 2D and 3D and between the cell lines used, which may help understand the observations in clinical samples. Finally, the authors link the gene expression of regulators of cytoskeletal dynamics to patient clinical outcome, demonstrating that RHOA, RAC1, LIMK1, and MYLK2 are associated with unfavourable prognosis.

This study shows cell behaviours that have a clinical relevance and may be comparable to other carcinomas. However, the study would benefit from a better connection between clinical sample analysis and the in vitro results. Furthermore, the bioinformatic analysis used bulk sequencing, and thus, the results cannot be attributed solely to the malignant compartment of the tumour but could be explained by increased stromal/fibrotic response observed in aggressive tumours, which the authors do not consider.

Comments:

1. Several of the protein inhibition results could be explained by the distinct protein expressions shown in Fig. 2B, however, the authors do not link these results in the text. For instance, the lower effect of blebbistatin on wound closure in HepG2 cells compared to SNU-475 cells can be attributed to the lower basal p-MLC.

2. In scratch assay, the authors should consider the possibility of changes in cell area and in

proliferation, which might affect the percentage of wound closure.

3. The authors compare 2D and 3D results, using type I collagen matrix for the 3D culture. The authors show that the proteins that are important for lamellipodia formation in 2D are not necessarily important for collagen invasion in 3D. Understanding these differences is important as 3D invasion is closer to in vivo conditions. However, the authors do not explore or discuss sufficiently the differences observed. First, the authors could compare the activation of key proteins in 2D vs 3D. Second, the authors should discuss the difference between 2D and 3D migration, which possibly explains most of their observations.

4. Collagen remodelling is necessary for migration in dense collagen matrices. The authors investigate collagen densification as a measure of remodelling, and mostly attribute this remodelling to biomechanical forces. However, collagen remodelling also involves enzyme-induced proteolysis, which should be considered as a possible mechanism playing a role in this system.

5. The authors use TCGA data to investigate the prognostic value of regulators of cytoskeletal dynamics. The authors find a link between RHOA, RAC1, LIMK1, and MYLK2 expression and poor prognosis. However, the expression of these genes cannot be directly attributed to the malignant component of the tumours, and could result from increased activation of hepatic stellate cells or increased composition of other mesenchymal cells. The authors could investigate the distinct expression of these genes/proteins in the different tumour compartments and could also compare their expression in the two distinct tumour subtypes described in Fig 1.

6. In the discussion, the authors show new results from Fig S21-22. I recommend presenting these results in the results section. Furthermore, the rationale for investigating upstream regulators is not clear.

Minor points:

1. Figure 3E legend only mentions SNU-475 but not HepG2.

To the editor:

We appreciate the time and effort that you and the reviewers dedicated to providing feedback on our manuscript and are grateful for the insightful comments on and valuable improvements to our paper. We thank the reviewers for their comments and have addressed each comment point by point. Please see below, in red, for a point-by-point response to the reviewers' comments and concerns. All page numbers refer to the revised manuscript file with tracked changes.

Reviewers' Comments to the Authors

Reviewer 1:

While the authors show that 2 types of cells exist in liver tumors *in vivo*, it is unclear if these are related to the cell lines chosen for analysis by anything other than histology. Molecular analysis of these cell types *in vivo* (such as IF of specific markers also tested in the 2D cell lines) would allow a more direct comparison.

We agree with the reviewer and have performed E-cadherin IHC on our tissue samples (description included on page 13):

“We have performed E-cadherin IHC on our tissue samples as an epithelial cell indicator (Fig 1A, lower panels). Liver cancer cells surrounded by fibrotic capsules show higher levels of E-Cadherin while the stromally invasive liver cancer has less E-cadherin. This further indicates the presence of two distinct liver cancer subtypes with the confined liver cancer being more epithelial and the stromally invasive cancer being less epithelial.”

Overall, this IHC demonstrates distinct liver cancer subtypes *in vivo* and that, in conjunction with our immunostaining and blotting, our cell lines tested are representative of these subtypes, with HepG2 cells expressing high E-cadherin levels and SNU475 cells expression low E-cadherin levels.

The TCGA data clearly show a relationship between the chosen cytoskeletal regulators and presence or absence of liver cancer. However, this does not shed any light on the relationship between these regulators and invasive capabilities of human liver tumors. Without this information it is unclear how this data is relevant to the 2D/3D experiments performed throughout the manuscript. I disagree that these data "implicates cytoskeletal dysregulation as a marker of liver cancer progression" since different stages of liver cancer are not analyzed.

We agree with the reviewer and have changed the claim of our bioinformatic analyses on page 18 under the section “**TCGA analysis reveals Rho, Rac, LIM Kinase predict poor prognosis**”:

“The TCGA analysis shows that overexpression and subsequent dysregulation of these cytoskeletal machinery are associated with liver cancer. These datasets are generated from bulk RNA sequencing and, therefore, cannot differentiate between nor directly attribute these differences to tumor cells. However, these data do suggest that targeting these pathways in the liver cancer microenvironment may be a valid therapeutic strategy. ”

We have also included in the discussion limitations of our bioinformatic analysis as well as future studies to address these limitations (on page 21):

“Our bioinformatic analyses suggest correlation of some of the markers examined in this study to HCC. This is in alignment with previous bioinformatic studies which have shown correlation of cytoskeletal markers, such as LIMK1, to higher hazard ratio of death in liver cancer patients ⁶⁵. However, these data are generated from bulk RNA sequencing and cannot directly attribute these differences to malignant components of the tumor. For example, these differences could also be attributed to hepatic stellate cells, which are the main effectors of liver cirrhosis and may be linked to liver cancer ⁶⁶. Future studies can use IHC to determine if stromally invasive liver cancer cells in patients have cytoskeletal dysregulation. Furthermore, these studies should include staging information of these datasets to correlate this dysregulation with liver cancer progression.”

Minor points:

Fig 2B - it is unclear why the bottom panels have 4 lanes across and why the samples were run on separate gels, making a direct comparison difficult.

We have reorganized to more clearly show that these are independent blots shown in Figure 2B on page 32.

The text refers to Fig 5D, but this doesn't appear in the actual Figure.

We have included figure 5D in Figure 5 on page 38 and have referenced it appropriately in the text.

Reviewer 2:

1. The choice of actin-cytoskeleton targeting drugs is poorly justified. A significant number of different chemical compounds is known to modulate actin filament turnover and actomyosin contractility. There is no rational explanation/ justification for the selection of specific chemicals in the present study. It seems that the study largely probes different steps of the Rac1 small GTPase signaling and it would be better if the author reshape their Introduction to get it more focused on Rac1 and its roles in liver cancer dissemination. Likewise, if the focus of this paper on the broad Rho family of small GTPases, this should be better described.

We thank the reviewer for this helpful suggestion and have focused the introduction on the Rho family of small GTPases on page 4 in the Introduction:

“At the core of this interplay is the Rho family of small GTPases. They key members of this family include RhoA which controls actin contractility, Rac1 which controls lamellipodia formation, and Cdc42 which is involved in filopodia formation¹⁰.

Both Rac1 and RhoA have been implicated in invasive behavior of various tumor types including breast, lung, colon, and liver cancer. Dysregulation of activity of both Rac1 and RhoA has been linked to mesenchymal tumor movement and invasive phenotypes in cancer¹¹. In addition, their downstream effectors have also been implicated in generating invasive cell motility.”

2. Along the same lines, using Sorafenib or BAPTA does not fit well into the design of this study. It seems that Sorafenib was used as a positive control, as a known inhibitor of cancer cell motility, which does not add much to the study. Ca chelator, BAPTA, would be a reasonable choice if the authors investigate calcium sensitive mechanisms of actomyosin contractility that involve Ca- and calmodulin-dependent myosin light chain kinase (MLCK). In this scenario, they should add MLCK inhibitors to their panel.

We agree with the reviewer and have added ML-7 as an MLCK inhibitor to our drug panel which has been added to Figures 3-6 and Figures S3-5,8,15-19. We have documented these results in the Results section:

Page 14:

“To interrogate cytoskeletal machinery that is directly downstream of calcium signaling, we tested inhibition of the calcium dependent myosin light chain kinase (MLCK). We find a significant decrease in wound closure for both cell lines after MLCK inhibition (Fig 3B).

Page 15:

“MLCK inhibition via ML-7 treatment decreased cellular solidity and circularity in SNU-475 cells (Fig. 3D, left, Fig. S4D). Interestingly, ML-7 treatment decreased HepG2 spread area and subsequently caused an increase in cell circularity (Fig 3D, right, Fig. S5B).”

Page 17:

“Interestingly, ML-7 treatment caused SNU-475 invading cells to completely lose their protrusions (Fig. 5A, 8th column). In addition, ML-7 treatment caused a decrease in invasion as well as short range and long range collagen remodeling (Fig. 5B-D). Conversely, HepG2 spheroids did not react significantly to ML-7 treatment.”

3a. While using chemical inhibitors of the actin cytoskeleton is an acceptable experimental approach to probe cytoskeletal mechanisms of cell phenotype and functions, the authors should discuss study limitations, which is possible nonspecific off-target effects of these compounds. This is particularly relevant to the Rac inhibitor NCS23766.

We agree with the reviewer and have added a section in the discussion about the limitations of using chemical inhibitors on page 19 paragraph 1:

“One limitation of this study is the use of pharmacological agents to study these pathways. It is known that these drugs can have off target effectors which may also affect our results^{55,56,55,57}. Future studies could address this by performing knockdowns/ knockouts of members of this pathway.”

3b. Also, the authors should report if the inhibitors affect cell viability and cell proliferation, especially under conditions of long exposure (up to 5 days), which could affect interpretation of their results.

We agree with the reviewer and have added results from Ki67 staining of our spheroids after 5 days of culture on page 16 and Fig S11:

“To determine if long term culture of these spheroids with drugs would affect proliferation, we stained our spheroids with Ki67 after 5 days of culture and did not see any significant changes in expression (Fig. S11).”

4. Immunoblotting data in Figure 2B and 3E lack important controls, such expression of total MLC and cofilin.

We agree with the reviewer and agree that total MLC and cofilin should be included as controls. We tried this, but the antibodies we used for this were not effective in our hands. We moved forward because we did not feel it would change the interpretation of the results.

5. Extracellular matrix remodeling was used as one of functional readouts, but the presented reflectance microscopy images in Fig 2E and 5C should be quantified.

We agree with the reviewer’s comment and have developed a script to quantify collagen density and its method can be found on page 9:

“**Collagen Density Calculation**

To quantify the bulk collagen remodeling by spheroids to their surrounding environment, we calculated the collagen density fold change as a function of distance from the spheroid. Each day-5 spheroid was first manually outlined in the brightfield channel, and then an average Z projection was taken in the reflectance channel. Using the trace and reflectance image, we calculated the centroid and drew radial line segments that were angled $\pi/6$ rads or 15° from each other. The line segments were long enough to reach a distant region with collagen intensity comparable to the background. For each radial line, we took plot profiles distanced $30\ \mu\text{m}$ from the edge of the spheroid and then averaged the plot profiles to obtain average intensity versus distance. In the averaged profile, we took the maximum intensity to be the first point of each profile and the plateau intensity to be the average of the last 30 points of the profile. We then divided every value by the plateau intensity to get collagen density fold change. The short range collagen density fold change is defined as the collagen density fold change $30\ \mu\text{m}$ away from the

spheroid (the first measurement of the average collagen density fold change). The long range intensity fold change is defined as the collagen density fold change 80 μm away from the edge. We note here that we used the confocal reflectance signal as a label-free relative measure of collagen density, as collagen has a high reflectance signal, but reflectance is not specific for collagen alone. The original outline of the spheroid was manually traced in ImageJ while the rest of the calculations were performed in Python 3.”

The results of this can be seen in Fig. 2F (lower right), Fig. 5D-E, and Fig. S15.

6. The choice of targets for the correlation analysis presented in Figure 7 is not optimal. MYLK2 does not really fit into Rho-GTPases pathways and was not investigated in the experimental part of this paper. Why only LIMK1 isoform was included? What about LIMK2 expression? Since the authors examined RhoA expression it would be logical to include immediate downstream effectors of RhoA, such as ROCK1 and ROCK2. Why PAK4 was omitted from the analysis since PAK4 inhibition was extensively probed in the experimental part of the paper.

We have included LIMK2, ROCK1, ROCK2, and PAK4 to our bioinformatic analysis in Figure 7 and Fig S21.

7. The note that actin targeting actin filament turnover and actomyosin contractility may have different effects on cell migration and invasion under 2-D and 3-D conditions is not novel and has been previously described for other cancer types (see for example PMC2630553 and PMC7916823). This should be properly discussed and cited.

We thank the reviewer for this suggestion and added the following paragraph to our discussion:

“While much of the same machinery in 2D migration is conserved in 3D migration, the presentation of the machinery in cells can vary greatly. In 2D migration, cells undergo repetitive lamellipodial protrusions, adhesion, and backside contraction⁶. In 3D migration, there are many additional factors cells face including migrating through subcellular-scaled pores, confining environments, and matrix degradation⁴⁸. Cells in 3D are able to also generate protrusions- though typically smaller/thinner in scale compared to their 2D counterparts- into the surrounding matrix. These protrusions have a wide variety of roles: mechanically probing the environment, applying contractile forces to remodel the surrounding matrix, anchoring matrix proteases to the cell surface, and generating migratory forces^{49, 50}. The ability of these protrusions to extend out and generate these forces is largely regulated by actomyosin contractility and actin turnover⁵¹. While loss of these protrusions via inhibition of actomyosin contractility greatly mitigate invasive properties of cells in 2D, this is not necessarily true in 3D migration. Cells have other modes of migration which are unique to 3D culture such as bleb based amoeboid migration which typically does not require protrusions^{52,53}. Actomyosin regulation in 3D has been extensively studied in other cancer types, such as breast and colorectal cancer^{17,5254}. In the context of liver cancer, while some chemical inhibitors may be able to stop invasive phenotypes on 2D surfaces, they may not be effective in stopping 3D migration. Thus, being able to differentiate between the effects of certain cytoskeletal inhibitors in distinct liver cancer types

requires more physiologically relevant 3D culture to capture the complexities of additional migration modes.”

8. Discussion is shallow and largely repeats the Result section. The authors should put more in depth discussion of how coordinated actin filament turnover and actomyosin contractility drives liver cancer cell invasion and how these mechanisms could be different under 2-D and 3-D environment.

We agree with the reviewer and have added more in depth discussion of the role of actin turnover and actomyosin contractility in 2D vs 3D migration for all drugs tested. These points can be found in the discussion:

Page 21:

“We found that inhibiting cofilin phosphorylation via LIMK1 inhibition was able to stifle 2D cell migration and reduce actin fiber thickness but was not able to decrease 3D spheroid invasiveness in both cell lines. Similarly, it has previously been reported that inhibition of LIMK1 weakens actin stability and collective cell migration but does not significantly affect single cell invasion⁶³. This may explain why our results do not show decreased invasion of SNU-475 cells after LIMK1 inhibition.”

Page 21:

“These thin protrusions were not able to induce significant collagen remodeling. This suggests that cells were able to probe their surrounding environment but were not tensile enough to produce contractile and migratory force. Cells treated with MLCK inhibitor, which also blocks actomyosin contractility, show decreased 3D spheroid invasion in SNU-475s. However, cells were more rounded with less prominent protrusions. Overall, this demonstrates that 3D cultures are able to distinguish between these distinct phenotypes arising from differential regulation of actomyosin contractility. These differences can arise from regulation of other components of this pathway such as ROCK and RhoA. Future experiments can help decouple the role of these components in 3D migration.“

Reviewer 3:

1. Several of the protein inhibition results could be explained by the distinct protein expressions shown in Fig. 2B, however, the authors do not link these results in the text. For instance, the lower effect of blebbistatin on wound closure in HepG2 cells compared to SNU-475 cells can be attributed to the lower basal p-MLC.

We agree with the reviewer and have included this in the discussions on page 20:

“Ultimately, these cells’ response to drugs is in part due to biochemical regulation: SNU-475 cells are more mesenchymal and have a greater amount of p-MLC than do HepG2 cells which makes the SNU-475 better candidates for studying cytoskeletal regulation.”

2. In scratch assay, the authors should consider the possibility of changes in cell area and in proliferation, which might affect the percentage of wound closure.

We agree with the reviewer and have added Ki67 staining of our spheroids after 5 days of culture on page 16 and Fig S11:

“To determine if long term culture of these spheroids with drugs would affect proliferation, we stained our spheroids with Ki67 after 5 days of culture and did not see any significant changes in expression (Fig. S11).”

In addition, we note the effect of proliferation on interpreting scratch assay results in the discussions and use it as rationale for 3D assays beginning on page 19:

“Our scratch assays show differential response to cytoskeletal inhibition between both cell lines and demonstrate clear differences in their dissemination ability (Fig 3B). However, the assay itself does not take into account key factors which would affect wound closure rate such as proliferation. In addition, the assay lacks the ability to capture certain modes of migration which can only be found in 3D assays. Thus, we turn to studying our cell lines in 3D collagen matrices.”

3a. The authors compare 2D and 3D results, using type I collagen matrix for the 3D culture. The authors show that the proteins that are important for lamellipodia formation in 2D are not necessarily important for collagen invasion in 3D. Understanding these differences is important as 3D invasion is closer to in vivo conditions. However, the authors do not explore or discuss sufficiently the differences observed. First, the authors could compare the activation of key proteins in 2D vs 3D.

We agree with the reviewer and have included 3D immunofluorescence of actin profiles in spheroids to better understand key cytoskeletal regulation differences in 2D vs 3D. This can be found in Fig 6B and is discussed on page 18:

“Cross sectional images of spheroids show distinct invasion profiles of SNU-475 cells based on different cytoskeletal perturbations (Fig 6B). While Rac1 inhibition was able to mitigate 3D invasion, invading cells had strong actin intensity indicating the presence of tensional actin stress fibers. In contrast, cells treated with blebbistatin and blebbistatin + LIMKi3 had much thinner protrusions, which explains why these cells performed less collagen remodeling. In the extreme case, treatment with MLCK or PAK4 inhibitors caused complete loss of protrusions with invaded cells adopting a rounder phenotype.”

3b. Second, the authors should discuss the difference between 2D and 3D migration, which possibly explains most of their observations.

We have also discussed the key protein activation differences in 2D vs 3D in the discussions on page 19:

“While much of the same machinery in 2D migration is conserved in 3D migration, the presentation of the machinery in cells can vary greatly. In 2D migration, cells undergo repetitive lamellipodial protrusions, adhesion, and backside contraction⁶. In 3D migration, there are many additional factors cells face including migrating through subcellular-scaled pores, confining environments, and matrix degradation⁴⁸. Cells in 3D are able to also generate protrusions- though typically smaller/thinner in scale compared to their 2D counterparts- into the surrounding matrix. These protrusions have a wide variety of roles: mechanically probing the environment, applying contractile forces to remodel the surrounding matrix, anchoring matrix proteases to the cell surface, and generating migratory forces^{49,50}. The ability of these protrusions to extend out and generate these forces is largely regulated by actomyosin contractility and actin turnover⁵¹. While loss of these protrusions via inhibition of actomyosin contractility greatly mitigate invasive properties of cells in 2D, this is not necessarily true in 3D migration. Cells have other modes of migration which are unique to 3D culture such as bleb based amoeboid migration which typically does not require protrusions^{52,53}. Actomyosin regulation in 3D has been extensively studied in other cancer types, such as breast and colorectal cancer^{17,5254}. In the context of liver cancer, while some chemical inhibitors may be able to stop invasive phenotypes on 2D surfaces, they may not be effective in stopping 3D migration. Thus, being able to differentiate between the effects of certain cytoskeletal inhibitors in distinct liver cancer types requires more physiologically relevant 3D culture to capture the complexities of additional migration modes.”

4. Collagen remodelling is necessary for migration in dense collagen matrices. The authors investigate collagen densification as a measure of remodelling, and mostly attribute this remodelling to biomechanical forces. However, collagen remodelling also involves enzyme-induced proteolysis, which should be considered as a possible mechanism playing a role in this system.

We agree with the reviewer and have included discussions about the role of enzyme induced proteolysis in this study:

“While our study focuses on the mechanical basis of collagen remodeling in 3D matrices, chemical signals such as MMPs which degrade collagen are also necessary for remodeling and invasion to occur⁶¹. Future studies will examine how altering degradation capabilities will affect differential invasive phenotypes in liver cancer.”

5. The authors use TCGA data to investigate the prognostic value of regulators of cytoskeletal dynamics. The authors find a link between RHOA, RAC1, LIMK1, and MYLK2 expression and poor prognosis. However, the expression of these genes cannot be directly attributed to the malignant component of the tumours, and could result from increased activation of hepatic stellate cells or increased composition of other mesenchymal cells. The authors could investigate the distinct expression of these genes/proteins in the different tumour compartments and could also compare their expression in the two distinct tumour subtypes described in Fig 1.

We agree with the reviewer and have changed the claim of our bioinformatic analyses on page 18 under the section **“TCGA analysis reveals Rho, Rac, LIM Kinase predict poor prognosis”**:

“The TCGA analysis shows that overexpression and subsequent dysregulation of these cytoskeletal machinery are associated with liver cancer. These datasets are generated from bulk RNA sequencing and, therefore, cannot differentiate between nor directly attribute these differences to tumor cells. However, these data do suggest that targeting these pathways in the liver cancer microenvironment may be a valid therapeutic strategy.”

We have also included in the discussion limitations of our bioinformatic analysis as well as future studies to address these limitations on page 21:

“Our bioinformatic analyses suggest correlation of some of the markers examined in this study to HCC. This is in alignment with previous bioinformatic studies which have shown correlation of cytoskeletal markers, such as LIMK1, to higher hazard ratio of death in liver cancer patients ⁶⁵. However, these data are generated from bulk RNA sequencing and cannot directly attribute these differences to malignant components of the tumor. For example, these differences could also be attributed to hepatic stellate cells, which are the main effectors of liver cirrhosis and may be linked to liver cancer ⁶⁶. Future studies can use IHC to determine if stromally invasive liver cancer cells in patients have cytoskeletal dysregulation. Furthermore, these studies should include staging information of these datasets to correlate this dysregulation with liver cancer progression.”

6. In the discussion, the authors show new results from Fig S21-22. I recommend presenting these results in the results section. Furthermore, the rationale for investigating upstream regulators is not clear.

We thank the reviewer for this helpful suggestion and have taken these figures out of the paper as not to focus on these upstream regulators.

Minor points:

1. Figure 3E legend only mentions SNU-475 but not HepG2.

We thank the reviewer for this helpful suggestion and have changed Figure 3E's caption accordingly on page 35:

“E) p-MLC western blots for drug conditions for SNU-475 and HepG2 cells. One-way ANOVA with Tukey post-hoc testing was performed. Significant difference ($p < 0.05$) was detected between any two of the conditions. $*P < 0.05$, $**P < 0.01$.”

REVIEWERS' COMMENTS:

Reviewer #1 (Remarks to the Author):

The authors were very responsive to my comments and have addressed all concerns. This is a solid manuscript that will be of interest to many in the field.

Reviewer #2 (Remarks to the Author):

The authors did a great job in addressing my major comments and questions. I have nothing to add.

Reviewer #3 (Remarks to the Author):

The authors successfully addressed all my points.

Minor comment:

1. The verb tense of some of the added sentences (in red) is not always consistent with the rest of the paragraph. For instance on page 16: "Measures of invasion area, farthest migration distance, percentage occupancy, and core area fold change **were** measured after 5 days of growth (Fig. 5B, Fig. S8). We **have also measured** changes in collagen densification as a result of drug treatment (Fig 5D-E, Fig. S9)."